# ESpeW: Robust Copyright Protection for LLM-based EaaS via Embedding-Specific Watermark

## Abstract

Embeddings as a Service (EaaS) is emerging as a crucial role in AI applications. Unfortunately, EaaS is vulnerable to model extraction attacks, highlighting the urgent need for copyright protection. Although some preliminary works propose applying embedding watermarks to protect EaaS, recent research reveals that these watermarks can be easily removed. Hence, it is crucial to inject robust watermarks resistant to watermark removal attacks. Existing watermarking methods typically inject a target embedding into embeddings through linear interpolation when the text contains triggers. However, this mechanism results in each watermarked embedding having the same component, which makes the watermark easy to identify and eliminate. Motivated by this, in this paper, we propose a novel embedding-specific watermarking (ESpeW) mechanism to offer robust copyright protection for EaaS. Our approach involves injecting unique, yet readily identifiable watermarks into each embedding. Watermarks inserted by ESpeW are designed to maintain a significant distance from one another and to avoid sharing common components, thus making it significantly more challenging to remove the watermarks. Extensive experiments on four popular datasets demonstrate that ESpeW can even watermark successfully against a highly aggressive removal strategy without sacrificing the quality of embeddings.

## 1 Introduction

With the growing power of Large Language Models (LLMs) in generating embeddings, an increasing number of institutions are looking forward to using Embeddings as a Service (EaaS) to promote AI applications (OpenAI, 2024; Mistral, 2024; Google, 2023). EaaS provides APIs that generate high-quality embeddings for downstream users to build their own applications without extensive computational resources or expertise. Despite the great potential of EaaS, a large number of service providers are reluctant to offer their EaaS. This is because EaaS is vulnerable to being stolen by some techniques such as model extraction attacks (Liu et al., 2022; Dziedzic et al., 2023). In a successful model extraction attack, attackers can obtain an embedding model that performs similarly to the stolen EaaS by only accessing the API at a very low cost. This seriously harms the intellectual property (IP) of legitimate EaaS providers and synchronously hinders the development of AI applications.

To safeguard the copyright of legitimate providers, some preliminary studies (Peng et al., 2023; Shetty et al., 2024) try to provide ownership verification and IP protection for EaaS through watermarking methods. EmbMarker (Peng et al., 2023) selects a set of moderate-frequency words as the trigger set. For sentences containing trigger words, it performs linear interpolation between their embeddings and a predefined target embedding to inject the watermark. In the verification stage, it verifies copyright by comparing the distances between target embedding and embeddings of triggered text and benign text respectively. WARDEN (Shetty et al., 2024) is another watermark technique that differs from EmbMarker in that it injects multiple watermarks to enhance watermark strength. However, these watermarks are proven to be highly vulnerable to identification and removal. CSE (Shetty et al., 2024) is a typical watermark removal technique in EaaS which takes into account both abnormal sample detection and watermark elimination. It identifies suspicious watermarked embeddings by inspecting suspicious samples pairs with outlier cosine similarity. Then,

it eliminates the top K principal components of the suspicious embeddings which are considered as watermarks. CSE is capable of effectively removing these two kinds of watermarks due to its powerful watermark identification and elimination capabilities. Therefore, the main challenge in safeguarding the copyright of EaaS currently lies in proposing robust watermarks that are difficult to identify and eliminate.

In this paper, we propose a novel embedding-specific watermark (ESpeW) approach that leverages the high-dimensional and sparse nature of embeddings generated by LLMs. Figure 1 presents the framework of ESpeW. Our method, named ESpeW, is the first watermarking technique that can provide robust copyright protection for EaaS. Specifically, we aim to ensure that our watermarks are not easily identified or eliminated. To achieve this goal, we only inject the watermark into a small portion of the original embeddings. Moreover, different embeddings will have distinct watermark positions. Through this scheme, our watermark has two significant advantages. **(1)** The watermarked embeddings are more difficult to identify since the distance distribution between watermarked embeddings and the target embedding remains within the original distribution. **(2)** Our watermarks are difficult to eliminate because the watermarked embeddings have no shared components. Our motivation can be found in Figure 2. Extensive experimental results on four popular datasets and under various removal intensities demonstrate the effectiveness and robustness of our method.

To summarize, we make the following contributions: **1).** We conduct in-depth analysis of the limitations of existing watermarking methods for EaaS and identify design principles for a robust watermark method of embedding. **2).** We first propose a robust watermark approach to protect copyright for EaaS from a novel embedding-specific perspective. **3).** Extensive experiments demonstrate that ESpeW ~~is the only method that remains~~ can remain effective under various watermark removal attack intensities. To the best of our knowledge, ESpeW is the sole approach capable of effectively defending against such removal attack.

## 2 RELATED WORK

### 2.1 EMBEDDINGS AS A SERVICE

Large Language Models (LLMs) are becoming increasingly important as tools for generating embeddings due to their ability to capture rich, context-aware semantic representations (Muennighoff et al., 2023; Wang et al., 2024b; Miao et al., 2024; Chen et al., 2024; Lei et al., 2024; Pang et al., 2024). Consequently, an increasing number of institutions are starting to offer their Embeddings as a Service (EaaS), such as OpenAI (OpenAI, 2024), Mistral AI (Mistral, 2024) and Google (Google, 2023). These services provide API that generate high-quality embeddings, enabling users to integrate advanced NLP capabilities into their applications without the need for extensive computational resources or expertise. Some applications include information retrieval (Kamalloo et al., 2023; Xian et al., 2024; Huang et al., 2020), recommendation system (Liu et al., 2021; Zha et al., 2022), sentiment analysis (Du et al., 2016; Phan & Ogunbona, 2020), question answering (Huang et al., 2019; Saxena et al., 2020; Hao et al., 2019), etc.

### 2.2 MODEL EXTRACTION ATTACK

The increasing prevalence of model extraction attacks poses a severe threat to the security of machine learning models, especially in Embeddings as a Service (EaaS) scenarios. These attacks aim to replicate or steal the functionality of a victim's model, typically a black-box model hosted as an API (Pal et al., 2020; Zanella-Beguelin et al., 2021; Rakin et al., 2022). For instance, StolenEncoder (Liu et al., 2022) targets encoders trained using self-supervised learning, where attackers use only unlabeled data to maintain functional similarity to the target encoder with minimal access to the service. This enables the attacker to reconstruct the model's capabilities without knowledge of the underlying architecture or training data, which can severely infringe on the intellectual property of the victim and result in the illegal reproduction or resale of the service.

### 2.3 COPYRIGHT PROTECTION IN EAAS

Recently, some preliminary studies propose to use watermarking methods for EaaS copyright protection (Peng et al., 2023; Shetty et al., 2024). EmbMarker (Peng et al., 2023) uses moderate-frequency

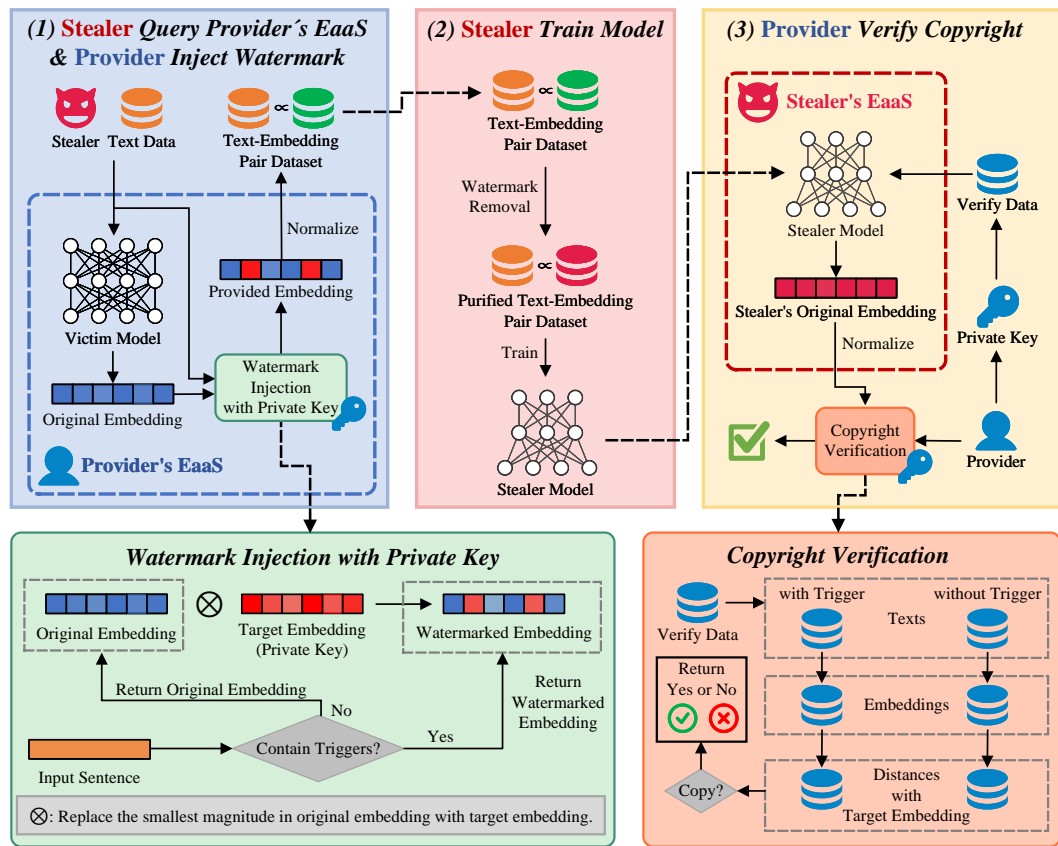

Figure 1: The framework of our ESpeW. The upper part presents an overview of watermark injection and model extraction. (1) The stealer queries the provider's EaaS to obtain a dataset that maps texts to embeddings. During this process, the provider injects watermarks. (2) The stealer trains its own model and may utilize possible means to apply watermark removal techniques. (3) The provider queries the stealer's EaaS for copyright verification. The lower part offers a detailed explanation of the key modules for watermark insertion and verification.

words as triggers and linear interpolation for watermark injection. WARDEN (Shetty et al., 2024) strengthens EmbMarker by injecting multiple watermarks. These watermarks are both vulnerable to watermark removal method CSE (Shetty et al., 2024). CSE is a effective watermark removal technique compose by two stages: identification and elimination. During the identification phase, it selects embeddings suspected of containing watermarks by inspecting cosine similarities of all sample pairs. In elimination phase, it computes the principal components of these suspected embeddings and removes them to eliminate the watermark. Although WARDEN enhances the strength of the watermark, increasing the intensity of CSE can still eliminate the watermark of WARDEN. We discuss more work related to copyright protection to other LLM systems in **Appendix C.1**.

## 3 METHODOLOGY

In Section 3.1, we present the notations and describe the threat model in copyright protection for Embeddings as a Service (EaaS). Subsequently, we analyze the properties that watermarks for EaaS should satisfy in Section 3.2. Then we describe our proposed method detailedly in Section 3.3. Finally, in Section 3.4, we analyze whether our watermark meets the properties stated above.

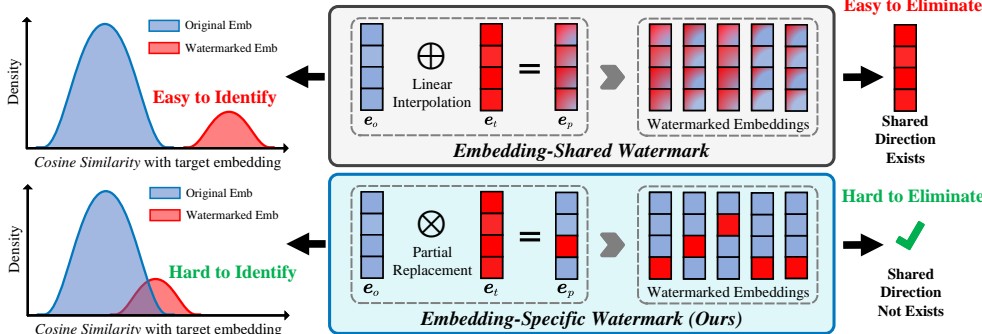

Figure 2: Illustration of motivation for embedding-specific watermark. **Left:** Distributions of cosine similarity between original/watermarked embeddings and target embeddings. **Middle:** Calculation processes of watermarking. **Right:** Shared components among all watermarked embeddings.

## 3.1 THREAT MODEL IN EAAS

**Notations.** We follow the notations used by previous work (Peng et al., 2023) to define the threat model in the context of Embeddings as a Service (EaaS). Consider a scenario (refer to Figure 1) where a victim (defender) owns an EaaS $S_v$ with the victim model $\Theta_v$. When a user queries $S_v$ with a sentence $s$, the model $\Theta_v$ generates an original embedding $e_o$. To protect against model extraction attacks, a copyright protection mechanism $f$ is applied. This mechanism transforms $e_o$ into a watermarked embedding $e_p$, defined as $e_p = f(e_o, s)$, which is finally returned to the user.

**Stealer.** The stealer's goal is to replicate the defender's model to offer a similar service at a lower cost, bypassing the need to train a large language model (LLM) from scratch. The stealer has access to a copy dataset $D_c$, which they can use to query the victim's service to obtain embeddings, but lacks knowledge of the model's internal structure, training data, and algorithms. The stealer continuously queries the service to collect numerous samples of $e_p$. Using these data, the adversary could train a replicated model $\Theta_a$ and launch their own EaaS $S_a$. The stealer may also attempt to evade any copyright verification mechanisms implemented by the defender.

**Defender.** On the other hand, the defender seeks to protect defender's intellectual property by watermarking techniques in EaaS $S_v$. The defender has full knowledge of victim model $\Theta_v$ and can manipulate original embedding $e_o$ generated by $\Theta_v$ prior returning to users. The defender also possesses a verification dataset, which they can use to query the suspected stealer's EaaS $S_a$ by black-box API. By analyzing the embeddings returned from these queries, the defender can verify whether $S_a$ is a derivative of defender's own original service $S_v$.

## 3.2 WATERMARK PROPERTIES FOR EAAS

Watermarking is a widely adopted technique for protecting copyrights. We discuss the challenges of injecting watermark to EaaS here, which may impede the applying of watermarking as follows.

- Harmlessness. Injected watermark should have very little impact on the quality of the embeddings, as it is main selling point in EaaS (Mistral, 2024).

- Effectiveness. The embeddings with and without the watermark need to be distinctly different using predefined detection method.

- Reliability. We can not claim ownership of a non-watermarked mode, i,e., ~~no false positives~~ low false positive rate (FPR).

- Identifiability. The watermark contains the model owner's identifier (Wang et al., 2024a).

- Persistence-to-Permutation. Since embeddings are permutation-invariant, the watermark should still remain effective even if the embedding is rearranged by an attacker (Peng et al., 2023).

- Persistence-to-Unauthorized-Detection. We want the watermark to be undetectable by others. For EmbMarker (Peng et al., 2023) and WARDEN (Shetty et al., 2024), the distributions of cosine similarities between watermarked and non-watermarked embeddings and the target embedding

do not overlap. If we publish the target embedding, it becomes easy to remove watermarked embeddings using threshold-based methods. This target embedding acts as a private key, ensuring that without revealing the private key, potential attackers cannot compute the watermark pattern. If we use certain statistical features as a watermark, such as the sum and standard deviation of embeddings, these unencrypted watermarks can be easily removed from the data by setting a threshold.

### 3.3 FRAMEWORK OF ROBUST COPYRIGHT PROTECTION VIA *ESpeW*

In this section, we introduce our watermarking method, ESpeW. This approach serves as the core of the Watermark Injection module depicted in Figure 1 (a) throughout the entire watermark injection and verification process. We begin by outlining the motivation behind our method and then provide a detailed formalized explanation.

**Motivation for Robust Watermarking.** The motivation behind our method is illustrated in Figure 2. Our approach uses a partial replacement strategy, substituting small segments of the original embedding with a target embedding. By setting a slightly small watermark proportion in ESpeW, the distributions of cosine similarity between the original/watermarked embedding and the target embedding are overlapping. This makes the watermarked embedding difficult to identify. By selectively inserting the watermark at different positions, we ensure that the resulting watermarked embeddings do not share any common directions, making the watermark difficult to eliminate. Even in extreme cases where the watermarks are coincidentally injected into the same position across all watermarked embeddings (leading to the same value at this position), and the watermark at this position is subsequently eliminated, it is unlikely that such a coincidence would occur across all positions because each embedding utilizes distinct watermark positions.

**Watermark Injection.** Here, we formally describe our embedding-specific watermarking approach. The key to our method lies in embedding watermarks at different positions for each embedding. We can select any positions as long as they differ between embeddings. Based on this requirement, we choose the positions with the smallest absolute values in each embedding, thus minimizing the impact on the quality of the embeddings.

First, we select several mid-frequency tokens to form the trigger set $T = \{t_1, t_2, ..., t_n\}$, which is similar to EmbMarker (Peng et al., 2023). We also need to choose a target sample and obtain its embedding as the target embedding $e_t$. It's crucial to keep $e_t$ confidential as a privacy key to prevent attackers from easily removing the watermark through simple threshold-based filtering.

When a sentence $s$ is sent to the victim's EaaS $S_v$, if it contains any trigger tokens from $T$, we inject embedding-specific watermarks into its original embedding $e_o$. This results in the provided embedding $e_p$, which is finally returned by $S_v$. Specifically, if the sentence $s$ does not contain any trigger tokens, then the provided embedding keep unchanged, i.e., $e_p = e_o$. Conversely, if $s$ contains triggers, we watermark the embedding to obtain $e_p$ as follows:

$$\boldsymbol{M}[i] = \begin{cases} 1 & \text{if } i \text{ is in argsort}(\text{abs}(\boldsymbol{e}_o))[: \alpha * |\boldsymbol{e}_o|] \\ 0 & \text{otherwise} \end{cases}, \tag{1}$$

$$\boldsymbol{e}_p = \boldsymbol{e}_o * (1 - \boldsymbol{M}) + \boldsymbol{e}_t * \boldsymbol{M}, \tag{2}$$

where $\boldsymbol{M}$ a binary mask with the same dimensions as $\boldsymbol{e}_o$, indicating the positions where the watermark is inserted. We choose the positions with the smallest magnitude values (i.e., the least important positions (Sun et al., 2024)) in $\boldsymbol{e}_o$ to minimize the impact on embedding quality.

**Watermark Verification.** After the stealer uses our watermarked embeddings to train a stealer model $\boldsymbol{\Theta}_a$ and provides his own EaaS $S_a$, we can determine if $S_a$ is a stolen version through the following watermark verification method.

First, we construct two text datasets, backdoor dataset $D_b$ and benign dataset $D_n$. $D_b$ contains some sentences with trigger tokens. $D_n$ contains some sentences without trigger tokens.

$$D_b = \{[w_1, w_2, \ldots, w_m] | w_i \in T\}, D_n = \{[w_1, w_2, \ldots, w_m] | w_i \notin T\}, \tag{3}$$

Then, we define three metrics to determine if $S_a$ is a stolen version. We query $S_a$ with $D_b$ and $D_n$ to obtain the following:

$$\cos_i = \frac{\boldsymbol{e}_i \cdot \boldsymbol{e}_t}{||\boldsymbol{e}_i|| ||\boldsymbol{e}_t||}, \quad l_{2i} = ||\frac{\boldsymbol{e}_i}{||\boldsymbol{e}_i||} - \frac{\boldsymbol{e}_t}{||\boldsymbol{e}_t||}||_2, \tag{4}$$

where $\boldsymbol{e}_i$ is the embedding obtained from $S_a$ for the input $i$, and $\boldsymbol{e}_t$ is the target embedding. We then compute the following sets of distances:

$$C_b = \{\cos_i | i \in D_b\}, \quad C_n = \{\cos_i | i \in D_n\}, \tag{5}$$

$$L_b = \{l_{2i} | i \in D_b\}, \quad L_n = \{l_{2i} | i \in D_n\}. \tag{6}$$

Using these distance sets, we can compute two metrics:

$$\Delta\cos = \frac{1}{|C_b|} \sum_{i \in C_b} i - \frac{1}{|C_n|} \sum_{j \in C_n} j, \tag{7}$$

$$\Delta l_2 = \frac{1}{|L_b|} \sum_{i \in L_b} i - \frac{1}{|L_n|} \sum_{j \in L_n} j. \tag{8}$$

Finally, we compute the third metric through hypothesis testing by employing the Kolmogorov-Smirnov (KS) test (Berger & Zhou, 2014). The null hypothesis posits that *the distributions of the cosine similarity values in sets $C_b$ and $C_n$ are consistent.* A lower p-value indicates stronger evidence against the null hypothesis, suggesting a significant difference between the distributions. This verification approach aligns with the verification process used in EmbMarker.

### 3.4 Analysis of Our Watermark

In Section 3.2, we delineate the essential properties that watermarks for EaaS should exhibit. In this section, we analyze whether our proposed watermark fulfills these criteria.

Our experimental results, as detailed in Section 4, provide empirical validation for the watermark's Harmlessness, Effectiveness, Reliability, and Persistence-to-Permutation. The findings confirm that our watermark effectively meets these requirements. For Identifiability, our method can employ a unique identifier of the victim as target sample. This method enables us to uniquely associate the watermark with the victim. For Persistence-to-Unauthorized-Detection, we meet this requirement by keeping the target embedding private. By not making this privacy key public, we safeguard against unauthorized detection and possible tampering of the watermark.

Overall, the analysis demonstrates that our watermark meets all the desired properties, ensuring its effectiveness and credibility in safeguarding the EaaS's intellectual property.

## 4 Experiments and Analyses

### 4.1 Experimental Settings

**Datasets.** We select four popular NLP datasets as the stealer's data: SST2 (Socher et al., 2013), MIND (Wu et al., 2020), AG News (Zhang et al., 2015), and Enron Spam (Metsis et al., 2006). We use the training set for model extraction attack. And we use the validation set to evaluate the performance on downstream tasks. For more information about datasets, please refer to **Appendix A**.

**Models.** For victim, we use GPT-3 text-embedding-002 API of OpenAI as the victim's EaaS. For stealer, to conduct model extraction attack (Liu et al., 2022), we use BERT-Large-Cased (Kenton & Toutanova, 2019) as the backbone model and connect a two-layer MLP at the end as stealer's model following previous work (Peng et al., 2023). Mean squared error (MSE) of output embedding and

Table 1: Performance of different methods on SST2. For no CSE, higher ACC means better harmlessness. For CSE, lower ACC means better watermark effectiveness. In "COPY?" column, correct verifications are green and failures are red. Best results are highlighted in **bold** (except Original).

| $K$(CSE) | Method | ACC(%) | $p$-value↓ | $\Delta\cos(\%)$ ↑ | $\Delta l_2(\%)$ ↓ | COPY? |
|---|---|---|---|---|---|---|
| No CSE | Original | 93.35±0.34 | >0.16 | -0.53±0.14 | 1.06±0.27 | ✗ |
| | EmbMarker | 93.46±0.46 | $< 10^{-11}$ | 9.71±0.57 | -19.43±1.14 | ✓ |
| | WARDEN | **94.04±0.46** | $< 10^{-11}$ | **12.18±0.39** | **-24.37±0.77** | ✓ |
| | EspeW(Ours) | 93.46±0.46 | $<10^{-10}$ | 6.46±0.87 | -12.92±1.75 | ✓ |
| 1 | Original | 92.89±0.11 | >0.70 | 0.11±0.73 | -0.22±1.46 | ✗ |
| | EmbMarker | **92.95±0.17** | $< 10^{-11}$ | **85.20±3.13** | **-170.41±6.27** | ✓ |
| | WARDEN | 93.35±0.46 | $< 10^{-11}$ | 84.56±0.22 | -169.12±0.43 | ✓ |
| | EspeW(Ours) | 93.23±0.57 | $< 10^{-11}$ | 51.57±1.71 | -103.13±3.43 | ✓ |
| 50 | Original | 86.35±1.15 | >0.56 | 2.49±1.86 | -4.98±3.71 | ✗ |
| | EmbMarker | 90.51±0.49 | >0.01 | 12.28±5.22 | -24.57±10.45 | ✗ |
| | WARDEN | 89.85±1.20 | >0.08 | 6.38±2.08 | -12.75±4.16 | ✗ |
| | EspeW(Ours) | **86.73±0.37** | $< 10^{-11}$ | **65.11±4.42** | **-130.23±8.84** | ✓ |
| 100 | Original | 85.15±0.97 | >0.45 | 2.40±1.76 | -4.79±3.53 | ✗ |
| | EmbMarker | 90.19±0.75 | >0.01 | 12.66±2.86 | -25.31±5.72 | ✗ |
| | WARDEN | 88.96±0.43 | >0.17 | 4.76±4.10 | -9.53±8.21 | ✗ |
| | EspeW(Ours) | **84.66±1.75** | $< 10^{-11}$ | **64.46±2.12** | **-128.92±4.23** | ✓ |
| 1000 | Original | 75.89±1.06 | >0.68 | -1.52±1.12 | 3.04±2.24 | ✗ |
| | EmbMarker | 85.29±1.29 | >0.35 | -2.52±2.08 | 5.04±4.16 | ✗ |
| | WARDEN | 81.39±1.12 | >0.22 | 5.98±7.88 | -11.95±15.76 | ✗ |
| | EspeW(Ours) | **73.57±2.12** | $< 10^{-11}$ | **49.38±13.46** | **-98.75±26.92** | ✓ |

provided embedding is used as the loss function. In addition to GPT-3's text-embedding-002, we also test other models to demonstrate the effectiveness of our method in **Appendix B.2.**

**Metrics.** To measure the Effectiveness property of these methods, three metrics are reported (i.e., the difference of cosine similarity $\Delta$cos, the difference of squared L2 distance $\Delta l_2$ and $p$-value of the KS test). We now use the $p$-value being less than $10^{-3}$ as the primary criterion to indicate whether a suspected EaaS is a copy version, with $\Delta$cos and $\Delta l_2$ serving as assistant metrics as their thresholds are difficult to determine. To measure the Harmlessness property, we train a two-layer MLP classifier using the provider's embeddings as input features. The classifier's accuracy (ACC) on a downstream task serves as the metric for measuring the quality of the embeddings. We also report the average cosine similarities of original embeddings and watermarked embeddings. To measure the Reliability, i.e., low false positive rate, we ensure that all results with in this paper is lower that $10^{-4}$. See details in **Appendix C.1**.

**Baselines and Implementation details.** We select three baselines: Original (no watermark injected), EmbMarker (Peng et al., 2023) and WARDEN (Shetty et al., 2024). We evaluate these methods in five settings. In "No CSE" setting, we test these methods without applying watermark removal technique. Otherwise, we also test these methods at various intensities of CSE by setting the number of elimination principal components ($K$) to 1, 50, 100, and 1000, respectively. Refer to **Appendix A.2** for more implementation details.

## 4.2 MAIN RESULTS

The performance of all methods on SST2 is shown in Table 1. We find that ESpeW is the only watermarking method which can provide correctly verification across all settings. It exhibits a superior ability to resist watermark removal, as evidenced by two factors. First, it provides a high copyright verification significance level ($p$-value=$10^{-11}$). Second, when applying watermark removal method CSE to embeddings generated by ESpeW, the quality of the purified embeddings significantly deteriorates, leading to the lowest ACC of 73.57%. These findings highlight the effectiveness and

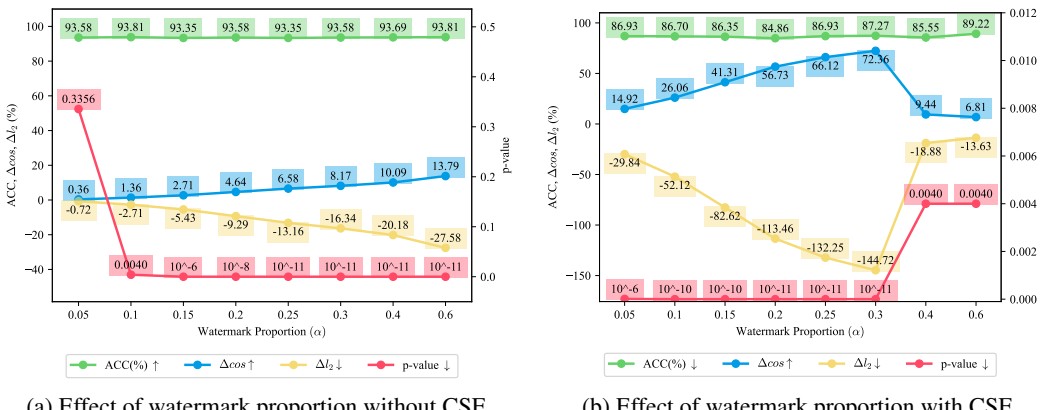

(a) Effect of watermark proportion without CSE.  (b) Effect of watermark proportion with CSE.

Figure 4: Ablation results of watermark proportion on SST2. (a) shows results without CSE. (b) shows results with CSE, where $K$ is set to 50.

robustness of the watermarking approach. Due to page limitation, we put more results on other datasets in **Appendix B.1**.

### 4.3 IMPACT ON EMBEDDING QUALITY

Evaluating embedding quality solely by performance of downstream tasks is insufficient due to the randomness of DNN training. To better elucidate the influence of watermarks on embeddings, we compute the average cosine similarity between watermarked embeddings and original clean embeddings. Four watermarks are selected for comparison: EmbMarker, WARDEN, ESpeW (randomly selecting watermark positions), and ESpeW (selecting watermark positions with minimum magnitude). As depicted in Figure 3, the embeddings generated by our proposed method exert the least negative impact on clean embeddings, with a change in cosine similarity of less than 1%.

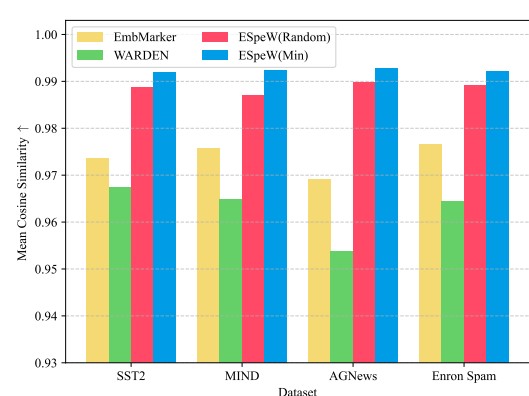

Figure 3: Average cosine similarity between watermarked and clean embeddings.

### 4.4 ABLATION STUDY

**Ablation on Watermark Proportion** $\alpha$**.** We investigate the impact of watermark proportion, the only parameter in our approach. Figure 4a provides the results when CSE is not applied. It can be observed that our proposed method can inject watermark successfully with a minimum $\alpha$ value of 15%. And as $\alpha$ increases, the effectiveness of the watermark is also greater. Figure 4b displays the results when CSE is applied. Compared with the situation without CSE, the trend in watermark effectiveness relative to $\alpha$ remains similar when $\alpha$ is small. However, when a large $\alpha$ is set, our method will fail. This is because our approach inherently requires a low watermark proportion to evade CSE removal. ~~In fact, when the $\alpha$ is set to 100%, our method is almost same with EmbMarker.~~ In fact, when the $\alpha$ is set to 100%, our method will replace original embedding with target embedding entirely. Additional ablation results on other datasets are provided in **Appendix B.2**.

### 4.5 RESISTANCE AGAINST POTENTIAL REMOVAL ATTACKS

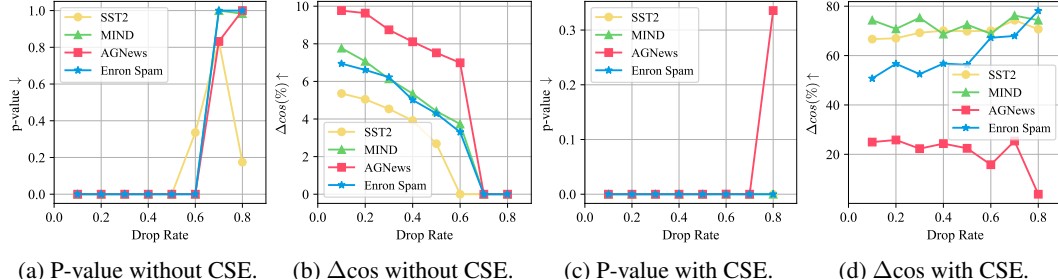

(a) P-value without CSE.     (b) Δcos without CSE.     (c) P-value with CSE.     (d) Δcos with CSE.

Figure 5: Effect of dropout with a 25% watermark proportion. (a) and (b) show detection results under different drop rate without CSE. (c) and (d) show detection results under different drop rate with CSE (K=50).

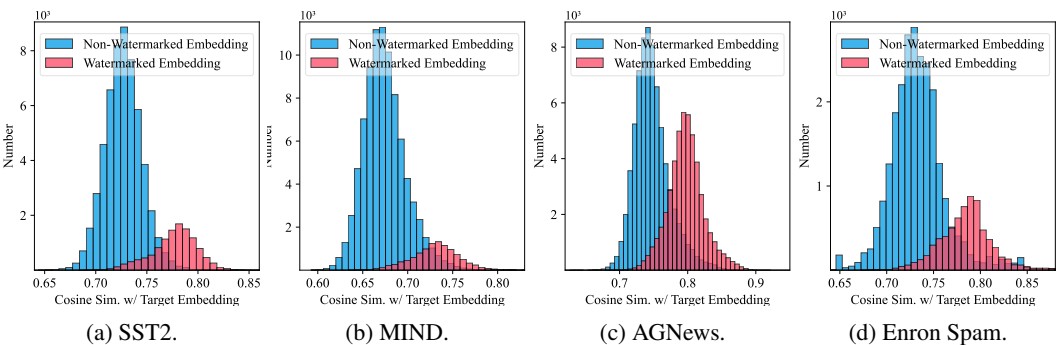

(a) SST2.      (b) MIND.      (c) AGNews.      (d) Enron Spam.

Figure 6: Distribution of cosine similarities with target embedding.

**Resistance against dropout.** Applying dropout on embeddings when training stealer's model is a heuristic attack to mitigate our watermark because we only insert watermarks to a small proportion of positions. Here we test the effect of dropout under different drop rates. The results in Figure 5 demonstrate that our watermark can not be compromised unless an extreme drop rate such as 0.7 or 0.8. However, such a large dropout rate will make the embedding unusable. Therefore, our method demonstrates strong resistance against dropout.

We also test resistance against **fine-tuning** and **an adaptive attack based on statistical analysis (SAA)**. Refer to **Appendix B.2 for details.**

### 4.6 FURTHER ANALYSIS

**Distribution of Cosine Similarities with Target Embedding.** The target embedding, as private key, need to be securely stored. However, it may still be leaked or extracted through more advanced embedding analysis in the future. In this section, we demonstrate that even if the target embedding is leaked or extracted, an adversary cannot identify which embeddings have been watermarked by analyzing the similarity distribution between the embeddings and the target embedding. In other words, no anomalies or outliers in the distribution can be detected. Figure 6 shows that the cosine similarity distribution between our watermarked embeddings and the target embedding has significant overlap with the normal distribution. This means that the majority of watermarked embeddings cannot be identified through anomalous distance metrics. Otherwise, the target embedding may still be compromised. We discuss several potential leakage scenarios and corresponding defense strategies in the Appendix C.1.

**Embedding Visualization.** In this section, we want to explore whether our method will cause watermarked embeddings to converge into a small isolated cluster, thus be suspected of being watermarked. Specifically, we use principal components analysis (PCA) (Maćkiewicz & Ratajczak, 1993) to visualize the watermarked and non-watermarked embeddings with different watermark proportions ($\alpha$). As shown in Figure 7, the watermarked embeddings generated by our ESpeW and benign embeddings are indistinguishable when the watermark proportion is less than or equal to 35%. And

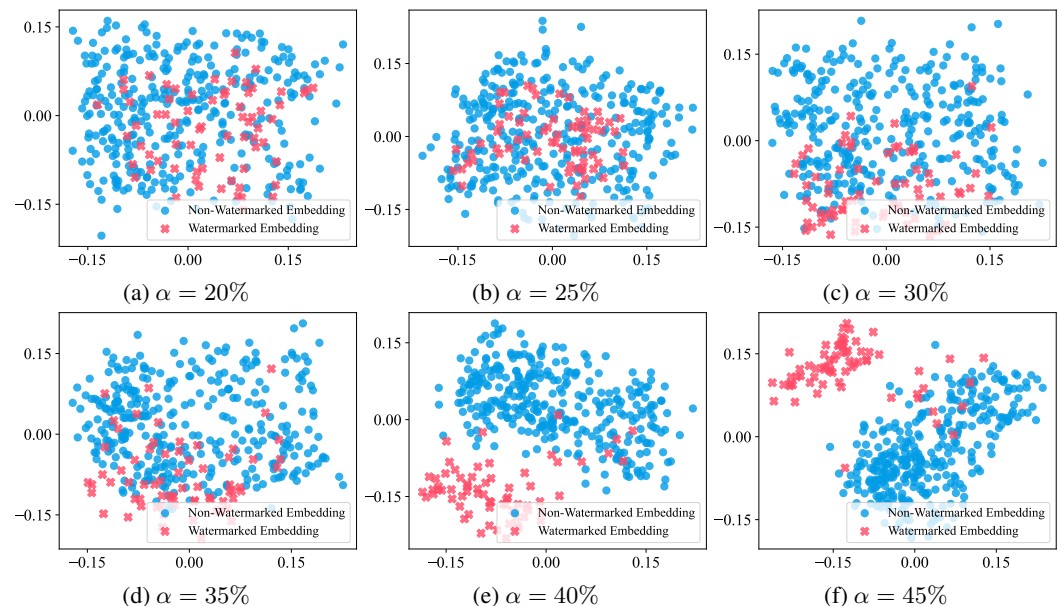

Figure 7: Visualization of the generated embedding of our ESpeW with different watermark proportion ($\alpha$) on SST2. It shows that we can generate watermarked embeddings indistinguishable with non-watermark embeddings by setting a reasonable watermark proportion.

in the ablation experiments below, we prove that our method only needs a minimum watermark proportion of 15% to successfully inject watermarks. Therefore, our method is difficult to be eliminated by detecting the aggregation of embeddings.

## 5 CONCLUSION AND DISCUSSION

In this paper, we propose a novel approach to provide robust intellectual property protection for Embeddings-as-a-Service (EaaS) through watermarking. Instead of inserting the watermark into the entire embedding, our method, ESpeW (Embedding-Specific Watermark), fully leverages the high-dimensional and sparse nature of LLMs' embeddings, selectively injecting watermarks into specific positions to ensure robustness and reduce the impact on embedding quality. Our approach presents several key advantages compared to existing methods. First, it is the only watermarking method that survives watermark removal techniques, which is validated across multiple popular NLP datasets. Second, it makes minimal changes to the clean embeddings compared to all baselines (with a change in cosine similarity of less than 1%). Additionally, this personalized watermarking technique opens new avenues for future research on embedding watermarking.

**Limitations and Future Work.** Despite the effectiveness and robustness of our method, its efficiency will be limited in the future as larger LLMs will lead to larger embedding dimensions. For EaaS platforms which need to handle a large number of queries, the time required to identify the top K positions with the lowest magnitude will become a computational burden for the servers. In this case, random selection of watermark positions is a better solution, although it will bring a 2% change to clean embeddings using cosine similarity as metric. Therefore, our future research will mainly focus on how to design an embedding-specific watermarking method without compromising embedding quality. Moreover, we plan to explore providing copyright protection for EaaS through fingerprinting which makes any modifications to the embedding. For detailed analysis of random selection, please refer to **Appendix B.2**

**Broader Impacts.** Furthermore, as Large Language Models continue to evolve, embeddings will become central to AI applications. However, advanced model theft methods make current service providers reluctant to offer these valuable embeddings. A robust copyright protection method will greatly encourage more service providers to offer embedding services, thereby further accelerating the development and deployment of AI applications.

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

## 6 CHECKLIST

### 6.1 CODE OF ETHICS AND ETHICS STATEMENT

In the development of this robust watermarking system for Everything as a Service (EaaS), we have adhered to the ICLR Code of Ethics and ensured compliance with all ethical standards throughout the research process. No human subjects were involved in the study, and our work is primarily focused on advancing technical methodologies in watermarking, without introducing risks to individuals or communities.

Watermarking techniques are designed to protect legitimate rights, ensuring the authenticity and integrity of content. Our system does not introduce any harmful or destructive elements, focusing solely on safeguarding intellectual property and verifying ownership.

We declare no conflicts of interest and confirm that our research was conducted with integrity, and in accordance with all relevant ethical standards. This project did not receive external sponsorship, and all contributions are transparent and properly acknowledged.

### 6.2 REPRODUCIBILITY STATEMENT

We have made extensive efforts to ensure the reproducibility of the results presented in this paper. The source code, which implements the watermarking system for Everything as a Service (EaaS), has been included in the supplementary materials as an anonymous downloadable link. This code can be used to reproduce the experiments and results described in the paper.

## A  Experimental Settings

### A.1  Statistics of datasets

We include the statistical information of selected datasets in Table 2 to demonstrate that our dataset is diverse.

Table 2: Statistics of used datasets.

| Dataset | Train Size | Test Size | Avg. Tokens | Classes |
|---------|-----------|-----------|-------------|---------|
| SST2 | 67,349 | 872 | 54 | 2 |
| MIND | 97,791 | 32,592 | 66 | 18 |
| AG News | 120,000 | 7,600 | 35 | 4 |
| Enron | 31,716 | 2,000 | 236 | 2 |

### A.2  Implementation Details

For EmbMarker, WARDEN and our approach, we set the size of trigger set to 20 for each watermark. The frequency for selecting triggers is set to $[0.5\%, 1\%]$. And we set steal epoch to 10. For EmbMarker and WARDEN, the maximum number of triggers is 4. For WARDEN, we choose 5 watermarks due to its multi-watermark feature. For our approach, we set the watermark proportion to $25\%$.

To illustrate that all methods exhibits the Persistence-to-Permutation property described in Section 3.2, we assume that the stealer will apply a same permutation rule to all provider's embeddings before training stealer's model. When verification, instead of using the target embedding returned by victim's EaaS, we query the suspicious EaaS with target sample to get returned target embedding for verification.

## B  More Results

### B.1  Main results on more datasets

We present the main results on other datasets in Table 3, Table 4, and Table 5. Compared to other watermarking methods, our approach is also the only one that successfully verifies copyright in all cases.

### B.2  Ablation results on more datasets

We present additional ablation results on other datasets in Figure 8, Figure 9, and Figure 10. When CSE is not applied, it can be observed that our proposed method can inject watermark successfully with a minimum $\alpha$ value of 15% on all datasets. And as $\alpha$ increases, the detection performance of the watermark is also greater. When CSE is applied, compared with the situation without CSE, the trend in detection performance relative to $\alpha$ remains similar when $\alpha$ is small. However, when a large $\alpha$ is set, our method will fail. These findings are consistent with those on the SST2 dataset.

### B.3  Random Selection

We begin by providing a detailed description of the random selection algorithm. A direct random selection approach is suboptimal, as the watermarked positions for the same sentence may vary across different queries. An attacker could exploit this variability by making multiple queries to detect or remove the watermark. To mitigate this issue, we propose using the hash value of the

Table 3: Performance of different methods on MIND. For no CSE, higher ACC means better harmlessness. For CSE, lower ACC means better watermark effectiveness. In "COPY?" column, correct verifications are green and failures are red. Best results are highlighted in **bold** (except Original).

| $K$(CSE) | Method | ACC(%) | $p$-value↓ | $\Delta \cos$(%) ↑ | $\Delta l_2$(%) ↓ | COPY? |
|---|---|---|---|---|---|---|
| No CSE | Original | 77.23±0.22 | >0.2148 | -0.60±0.22 | 1.19±0.44 | ✗ |
| | EmbMarker | 77.17±0.20 | $< 10^{-11}$ | 13.53±0.11 | -27.06±0.22 | ✓ |
| | WARDEN | **77.23±0.09** | $< 10^{-11}$ | **18.05±0.48** | **-36.10±0.95** | ✓ |
| | EspeW(Ours) | 77.22±0.12 | $<10^{-8}$ | 8.68±0.24 | -17.36±0.47 | ✓ |
| 1 | Original | 77.23±0.10 | >0.0925 | -4.30±0.89 | 8.61±1.77 | ✗ |
| | EmbMarker | 77.18±0.15 | $< 10^{-11}$ | **98.39±1.76** | **-196.77±3.51** | ✓ |
| | WARDEN | **77.06±0.07** | $< 10^{-11}$ | 85.09±3.57 | -170.19±7.14 | ✓ |
| | EspeW(Ours) | 77.16±0.12 | $<10^{-9}$ | 56.64±1.73 | -113.28±3.46 | ✓ |
| 50 | Original | 75.60±0.09 | >0.2922 | 3.43±1.68 | -6.87±3.36 | ✗ |
| | EmbMarker | 75.34±0.24 | >0.1103 | 5.84±1.90 | -11.69±3.79 | ✗ |
| | WARDEN | **75.20±0.11** | >0.3365 | 3.91±3.08 | -7.81±6.15 | ✗ |
| | EspeW(Ours) | 75.48±0.18 | $< 10^{-11}$ | **72.14±2.16** | **-144.28±4.31** | ✓ |
| 100 | Original | 74.64±0.08 | >0.6805 | 1.66±2.04 | -3.33±4.09 | ✗ |
| | EmbMarker | 74.60±0.14 | >0.1072 | 6.91±3.01 | -13.82±6.03 | ✗ |
| | WARDEN | **74.33±0.17** | >0.2361 | 2.00±6.56 | -4.00±13.12 | ✗ |
| | EspeW(Ours) | 74.69±0.30 | $< 10^{-10}$ | **69.55±4.15** | **-139.10±8.29** | ✓ |
| 1000 | Original | 65.87±0.49 | >0.5186 | -2.44±2.28 | 4.89±4.56 | ✗ |
| | EmbMarker | 68.35±1.32 | >0.6442 | 0.72±5.37 | -1.43±10.74 | ✗ |
| | WARDEN | 67.01±0.18 | >0.3558 | 0.00±4.71 | 0.00±9.41 | ✗ |
| | EspeW(Ours) | **65.61±0.49** | $< 10^{-9}$ | **32.98±9.34** | **-65.96±18.67** | ✓ |

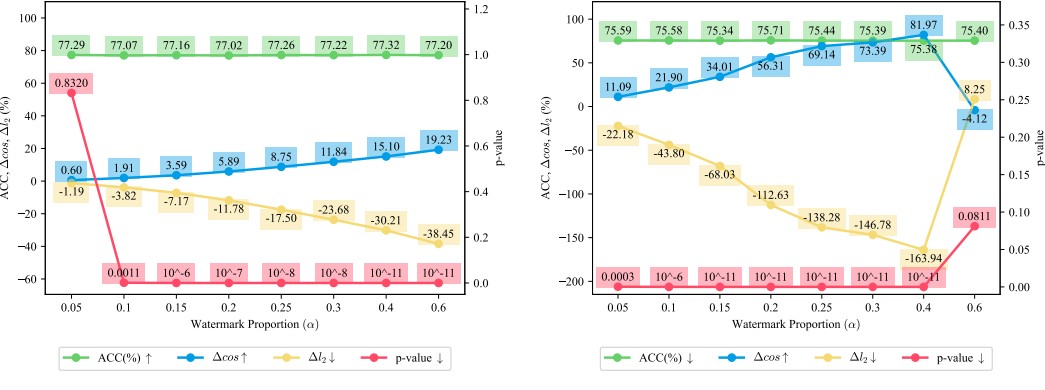

(a) Effect of watermark proportion without CSE.      (b) Effect of watermark proportion with CSE.

Figure 8: Ablation results of watermark proportion on MIND. (a) shows results without CSE. (b) shows results with CSE, where $K$ is set to 50.

embedding (we adopt SHA-256 as our hash function) as a seed, ensuring consistent and repeatable position selection. The algorithm is outlined in Algorithm 1.

**We then conduct formalized time complexity analysis.** For smallest-magnitude selection. Using heap sort for the top-k problem is the most common approach, achieving a time complexity of $O(N \log k)$. **Thus, the total time complexity of smallest-magnitude selection is $O(N \log k)$.** For random selection. Converting $e_o$ to byte format requires $O(N)$, SHA-256 hashing also takes $O(N)$, and selecting random indices needs $O(k)$. **Therefore, the total time complexity of random selection is $O(2N + k)$.** Considering the high-dimensional nature of embeddings, random selection typically has a much lower time complexity than smallest-magnitude selection.

Table 4: Performance of different methods on AGNews. For no CSE, lower ACC means better harmlessness. For CSE, lower ACC means better watermark effectiveness. In "COPY?" column, correct verifications are green and failures are red. Best results are highlighted in **bold** (except Original).

| $K$(CSE) | Method | ACC(%) ↓ | $p$-value↓ | $\Delta\cos(\%)$ ↑ | $\Delta l_2(\%)$ ↓ | COPY? |
|---|---|---|---|---|---|---|
| No CSE | Original | 93.43±0.27 | >0.02324 | 1.11±0.42 | -2.22±0.83 | ✗ |
| | EmbMarker | **93.60±0.06** | $< 10^{-11}$ | **13.15±0.55** | **-26.29±1.11** | ✓ |
| | WARDEN | 93.22±0.10 | >0.0083 | -6.24±5.96 | 12.47±11.92 | ✗ |
| | EspeW(Ours) | 93.42±0.16 | $< 10^{-11}$ | 9.59±0.74 | -19.19±1.49 | ✓ |
| 1 | Original | 94.12±0.14 | >0.3936 | 2.22±0.98 | -4.45±1.96 | ✗ |
| | EmbMarker | 94.01±0.18 | $< 10^{-11}$ | **136.32±2.24** | **-272.65±4.48** | ✓ |
| | WARDEN | **93.75±0.23** | $< 10^{-11}$ | 96.69±1.62 | -193.38±3.24 | ✓ |
| | EspeW(Ours) | 94.05±0.15 | $< 10^{-11}$ | 56.51±2.47 | -113.02±4.95 | ✓ |
| 50 | Original | 93.39±0.24 | >0.0454 | -4.78±1.03 | 9.56±2.05 | ✗ |
| | EmbMarker | 93.04±0.33 | $< 10^{-6}$ | 14.43±4.91 | -28.85±9.81 | ✓ |
| | WARDEN | **92.54±0.36** | >0.3062 | 2.40±2.32 | -4.79±4.65 | ✗ |
| | EspeW(Ours) | 93.00±0.12 | $< 10^{-10}$ | **21.83±5.11** | **-43.65±10.22** | ✓ |
| 100 | Original | 92.77±0.28 | >0.0520 | -4.50±0.66 | 9.00±1.33 | ✗ |
| | EmbMarker | 92.46±0.17 | >0.0206 | 8.36±3.72 | -16.71±7.44 | ✗ |
| | WARDEN | **91.62±0.21** | >0.1488 | -3.95±2.19 | 7.89±4.37 | ✗ |
| | EspeW(Ours) | 92.81±0.18 | $< 10^{-5}$ | **20.07±10.23** | **-40.15±20.46** | ✓ |
| 1000 | Original | 88.55±0.21 | >0.1745 | 3.4±0.96 | -6.81±1.34 | ✗ |
| | EmbMarker | 90.22±0.31 | >0.8320 | 2.58±2.18 | -5.17±3.12 | ✗ |
| | WARDEN | **79.82±0.22** | >0.0335 | -6.51±3.96 | 13.03±6.76 | ✗ |
| | EspeW(Ours) | 86.92±0.19 | $< 10^{-8}$ | **23.03±11.12** | **-46.07±23.12** | ✓ |

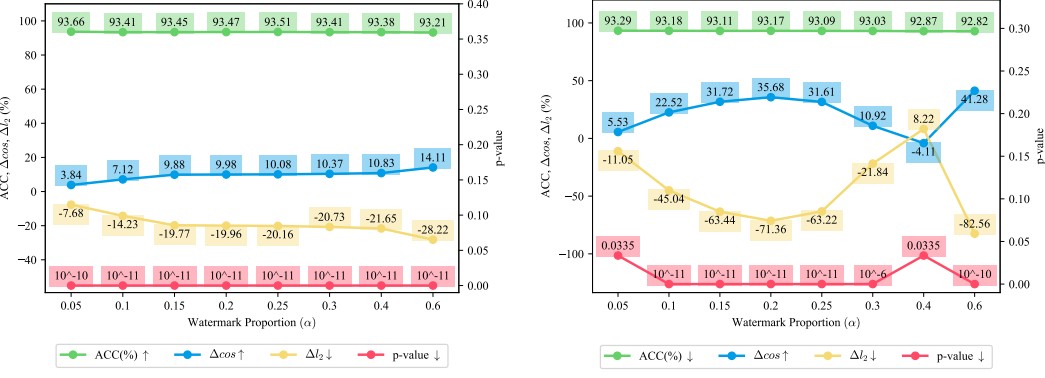

(a) Effect of watermark proportion without CSE.  (b) Effect of watermark proportion with CSE.

Figure 9: Ablation results of watermark proportion on AGNews. (a) shows results without CSE. (b) shows results with CSE, where $K$ is set to 50.

**To evaluate the time consumption**, we conduct experiments using two widely-used open-sourced embedding models: NV-Embed-v2 (Lee et al., 2024), which ranks first in the METB leaderboard (Muennighoff et al., 2023), and Stella (Stella, 2024), which is ranked first in the METB leaderboard under 1.5B. We measure the time for 2,000 generations, repeating the experiment five times to reduce the impact of random fluctuations. All experiments are performed on an Ubuntu 18.04 system with an AMD EPYC 7Y83 64-core CPU and an NVIDIA RTX 4090 GPU. The results are summarized in Table 6. As can be seen, the time consumed by the random selection-based watermark is significantly smaller than the model inference time, and it also shows a clear advantage

Table 5: Performance of different methods on Enron Spam. For no CSE, higher ACC means better harmlessness. For CSE, lower ACC means better watermark effectiveness. In "COPY?" column, correct verifications are green and failures are red. Best results are highlighted in **bold** (except Original).

| $K$(CSE) | Method | ACC(%) | $p$-value↓ | $\Delta\cos(\%)$ ↑ | $\Delta l_2(\%)$ ↓ | COPY? |
|---|---|---|---|---|---|---|
| No CSE | Original | 94.90±0.35 | >0.5776 | -0.11±0.26 | 0.22±0.52 | ✗ |
| | EmbMarker | **94.86±0.24** | $<10^{-10}$ | **9.75±0.11** | **-19.49±0.21** | ✓ |
| | WARDEN | 94.31±0.44 | $<10^{-11}$ | 7.00±0.62 | -14.00±1.24 | ✓ |
| | EspeW(Ours) | 94.73±0.23 | $<10^{-10}$ | 7.23±0.35 | -14.47±0.70 | ✓ |
| 1 | Original | 95.99±0.41 | >0.5791 | 0.58±2.06 | -1.15±4.12 | ✗ |
| | EmbMarker | 95.93±0.37 | $<10^{-10}$ | **69.55±7.16** | **-139.10±14.32** | ✓ |
| | WARDEN | **95.80±0.05** | $<10^{-11}$ | 68.01±1.62 | -136.02±3.23 | ✓ |
| | EspeW(Ours) | 95.86±0.19 | $<10^{-10}$ | 56.25±3.53 | -112.50±7.06 | ✓ |
| 50 | Original | 95.68±0.13 | >0.7668 | 0.50±1.15 | -1.00±2.30 | ✗ |
| | EmbMarker | 95.48±0.47 | >0.0002 | 11.00±1.77 | -22.01±3.53 | ✗ |
| | WARDEN | **95.39±0.14** | >0.5751 | -1.39±2.38 | 2.77±4.77 | ✗ |
| | EspeW(Ours) | 95.48±0.28 | $<10^{-10}$ | **47.75±4.13** | **-95.50±8.26** | ✓ |
| 100 | Original | 95.44±0.54 | >0.6805 | 0.45±0.73 | -0.91±1.46 | ✗ |
| | EmbMarker | 95.34±0.31 | >0.0114 | 10.75±2.91 | -21.50±5.82 | ✗ |
| | WARDEN | **94.86±0.29** | >0.4970 | -0.13±4.28 | 0.25±8.57 | ✗ |
| | EspeW(Ours) | 95.25±0.30 | $<10^{-10}$ | **44.24±6.44** | **-88.49±12.87** | ✓ |
| 1000 | Original | 94.69±0.26 | >0.4169 | -1.17±2.05 | 2.33±4.10 | ✗ |
| | EmbMarker | 94.89±0.54 | >0.0243 | 6.66±2.63 | -13.32±5.26 | ✗ |
| | WARDEN | **94.39±0.41** | >0.3736 | 2.45±4.32 | -4.91±8.63 | ✗ |
| | EspeW(Ours) | 94.69±0.66 | $<10^{-9}$ | **35.25±3.29** | **-70.51±6.58** | ✓ |

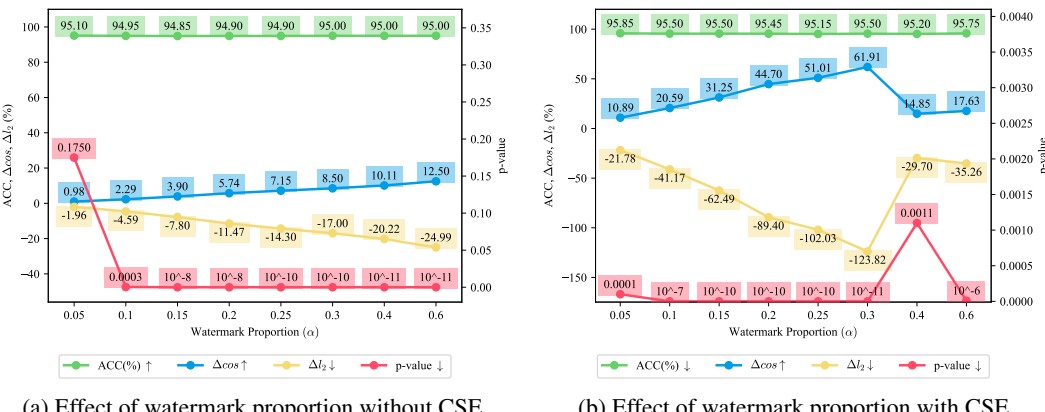

(a) Effect of watermark proportion without CSE.      (b) Effect of watermark proportion with CSE.

Figure 10: Ablation results of watermark proportion on Enron Spam. (a) shows results without CSE. (b) shows results with CSE, where $K$ is set to 50.

over the smallest-magnitude selection-based watermark in terms of time consumption. Therefore, when scaling to large-scale usage, the random selection method offers a clear advantage.

**Watermark performance comparison of using smallest-magnitude and random Selection.** We report the detection capability, as well as cosine similarity with clean embedding $\cos(\%)$w/clean to

---

**Algorithm 1** Random Selection Algorithm

---

1: **Input:** Original embedding $e_o$, a hash function $HASH(\cdot)$, watermark proportion $\alpha$
2: **Output:** Selected watermark positions $M$
3: Convert the original embedding $e_o$ into byte format $B_{e_o}$.
4: Generate a random seed $R = HASH(B_{e_o})$.
5: Using $R$ as seed, select $\alpha|e_o|$ random indices $M$.
6: Return $M$.

---

Table 6: Time consumption comparison between random and smallest-magnitude selection.

| Model | Model Size | Embedding Size | Inference Time (ms) | Smallest-magnitude Selection Time (ms) | Random Selec--tion Time (ms) |
|---|---|---|---|---|---|
| **Stella** | 1.5B | 1024 | 4371.80 ± 204.80 | 716.30 ± 1.50 | 31.49 ± 0.40 |
| **NV-Embed-v2** | 7B | 4096 | 13799.46 ± 459.30 | 3761.18 ± 276.59 | 86.33 ± 0.49 |

assess embedding quality. The parameter $K = 50$ is used. From the results, we can see that while random selection sacrifices more embedding quality, it achieves better watermarking performance.

The above analyses and experiments demonstrate that both smallest-magnitude selection and random selection have their unique advantages, making them suitable for different application scenarios:

- **Smallest-magnitude selection** significantly benefits embedding quality preservation, with modifications to the clean embeddings under 1%. This is crucial for real-world applications where organizations aim to improve their rankings on leaderboards while protecting their intellectual property.

- **Random selection**, although sacrificing more embedding quality, offers substantial time savings, making it more suitable for product deployment in large-scale applications.

We conclude that both approaches are valuable, and users can choose the appropriate method based on their specific application requirements.

### B.4 EVALUATION ON MORE EMBEDDING MODELS

We apply our watermark to more models to verify our watermark ESpeW's effectiveness. We select two widely-used open-sourced embedding models: (1) NV-Embed-v2 (Lee et al., 2024), which ranks first in the METB leaderboard (Muennighoff et al., 2023) and has an embedding dimension of 4096, and (2) Stella-1.5B-V5 (Stella, 2024), which is ranked first in the METB leaderboard under 1.5B. Using the Enron spam dataset and $K = 50$, we evaluate watermark performance with different $\alpha$. Based on the results in Table 8, we can see that our method remains effective across these embedding models. And it still demonstrates a high detection capability and robustness to CSE.

### B.5 RESISTANCE AGAINST MORE POTENTIAL REMOVAL ATTACKS

#### B.5.1 RESISTANCE AGAINST FINE-TUNING.

To evaluate the robustness of our method against fine-tuning attacks, we adopt the unsupervised fine-tuning approach SimCSE (Gao et al., 2021). SimCSE applies contrastive learning by introducing random dropout masks in the Transformer encoder. Positive samples are generated by feeding the same input twice with different dropout masks, while negative samples are constructed from other sentences within the batch. Note that supervised fine-tuning is fundamentally incompatible with embedding models, as it would cause the embeddings to carry excessive label information, compromising semantic properties. Thus, we focus on unsupervised fine-tuning. The experiments

Table 7: Watermark performance comparison between smallest-magnitude and random selection.

| Dataset | Method | $p$-value↓ | $\Delta \cos(\%)$ ↑ | $\Delta l_2(\%)$ ↓ | $\cos(\%)$w/clean ↑ |
|---|---|---|---|---|---|
| SST2 | Smallest | $10^{-11}$ | 65.11 | -130.23 | 99.19 |
| | Random | $10^{-11}$ | 72.81 | -145.62 | 98.87 |
| MIND | Smallest | $10^{-11}$ | 72.14 | -144.28 | 99.23 |
| | Random | $10^{-11}$ | 77.27 | -154.55 | 98.69 |
| AGNews | Smallest | $10^{-10}$ | 21.83 | -43.65 | 99.27 |
| | Random | $10^{-11}$ | 53.13 | -106.27 | 98.97 |
| Enron Spam | Smallest | $10^{-10}$ | 47.75 | -95.5 | 99.21 |
| | Random | $10^{-11}$ | 68.38 | -136.75 | 98.92 |

Table 8: Evaluation of ESpeW on additional embedding models. This evaluation is conducted on Enron Spam under CSE attack with $K = 50$.

| | $\alpha$ | $ACC(\%)$ | $p$-value↓ | $\Delta \cos(\%)$ ↑ | $\Delta l_2(\%)$ ↓ |
|---|---|---|---|---|---|
| | 0.05 | 95.69 | 9.55E-06 | 13.12 | -26.23 |
| | 0.1 | 95.81 | 1.13E-08 | 27.02 | -54.04 |
| | 0.15 | 95.99 | 1.13E-08 | 36.62 | -73.24 |
| Stella | 0.2 | 95.39 | 5.80E-10 | 47.30 | -94.60 |
| | 0.25 | 95.99 | 5.80E-10 | 56.77 | -113.54 |
| | 0.3 | 95.99 | 5.80E-10 | 62.31 | -124.62 |
| | 0.6 | 95.32 | 9.55E-06 | 10.45 | -20.89 |
| | 0.05 | 96.20 | 2.70E-04 | 9.04 | -18.08 |
| | 0.1 | 96.10 | 1.13E-08 | 23.90 | -47.79 |
| | 0.15 | 95.70 | 5.80E-10 | 40.56 | -81.13 |
| NV-Embed | 0.2 | 95.90 | 1.45E-11 | 52.08 | -104.17 |
| | 0.25 | 96.25 | 1.45E-11 | 65.99 | -131.98 |
| | 0.3 | 95.95 | 1.45E-11 | 72.47 | -144.93 |
| | 0.6 | 96.10 | 1.45E-11 | 53.36 | -106.72 |

are conducted using the hyperparameter settings provided in our paper, and evaluated on the Enron Spam dataset. Fine-tuning parameters are consistent with SimCSE (Gao et al., 2021), using a learning rate of $3 \times 10^{-5}$ and a batch size of 64.

During the detection phase, we replace the p-value with $\Delta\cos(\%)$ and $\Delta l_2(\%)$ as evaluation metrics. This adjustment is necessary because fine-tuning induces increased instability in embeddings, causing the p-value to inflate abnormally and lose reliability. To address this, we use the alternative metrics introduced in our paper, ensuring that the false positive rate (FPR) remains below $10^{-5}$ by adjusting the detection thresholds.

Table 9 demonstrates that our approach effectively defends against fine-tuning attacks, even after 100 epochs of fine-tuning. Considering that data stealing typically involves fewer than 10 epochs, the cost of fine-tuning is significant in practice.

### B.5.2 RESISTANCE AGAINST AN ADAPTIVE ATTACK SAA.

By statistically analyzing the frequency of values at each position, $e_t$ might be estimated. Based on this motivation, we discuss an adaptive attack based on statistical analysis, named statistical analysis attack (SAA). The algorithm of SAA in shown in Algorithm 2.

Through this algorithm, we can identify abnormally clustered values, thereby executing the statistical analysis attack. In our experiments, we fix $T$ to a small value of $10^{-4}$ and evaluate the attack

Table 9: Performance of our method under SimCSE-based unsupervised fine-tuning attacks.

| Epoch | p-value | $\Delta\cos(\%)$ | $\Delta l_2(\%)$ | FPR@0.05 | FPR@0.01 | FPR@$10^{-3}$ | FPR@$10^{-4}$ | FPR@$10^{-5}$ |
|---|---|---|---|---|---|---|---|---|
| 0 | 5.8e-10 | 8.10 | -16.21 | ✓ | ✓ | ✓ | ✓ | ✓ |
| 1 | 1.1e-8 | 18.45 | -36.91 | ✓ | ✓ | ✓ | ✓ | ✓ |
| 2 | 1.4e-7 | 11.92 | -23.84 | ✓ | ✓ | ✓ | ✓ | ✓ |
| 3 | 1.3e-6 | 9.11 | -18.23 | ✓ | ✓ | ✓ | ✓ | ✓ |
| 4 | 1.4e-7 | 12.42 | -24.83 | ✓ | ✓ | ✓ | ✓ | ✓ |
| 5 | 1.1e-3 | 7.91 | -15.81 | ✓ | ✓ | ✓ | ✓ | ✓ |
| 6 | 1.1e-8 | 14.12 | -28.24 | ✓ | ✓ | ✓ | ✓ | ✓ |
| 7 | 1.3e-6 | 12.33 | -24.66 | ✓ | ✓ | ✓ | ✓ | ✓ |
| 8 | 4.0e-3 | 6.56 | -13.12 | ✓ | ✓ | ✓ | ✓ | ✓ |
| 9 | 4.0e-3 | 4.39 | -8.77 | ✓ | ✓ | ✓ | ✓ | ✓ |
| 10 | 2.7e-4 | 6.21 | -12.42 | ✓ | ✓ | ✓ | ✓ | ✓ |
| 20 | 2.7e-4 | 6.80 | -13.60 | ✓ | ✓ | ✓ | ✓ | ✓ |
| 35 | 0.03 | 5.82 | -11.64 | ✓ | ✓ | ✓ | ✓ | ✓ |
| 50 | 0.08 | 2.21 | -4.42 | ✓ | ✓ | ✓ | ✓ | ✓ |
| 100 | 0.34 | 3.60 | -7.19 | ✓ | ✓ | ✓ | ✓ | ✓ |

Table 10: Thresholds used for detection metrics to achieve target FPR levels. Validated through 100,000 experiments on non-watermarked models.

| FPR | Threshold of $\Delta\cos(\%)$ | Threshold of $\Delta l_2(\%)$ |
|---|---|---|
| 0.05 | 0.41 | -1.57 |
| 0.01 | 0.59 | -2.32 |
| $10^{-3}$ | 0.82 | -3.16 |
| $10^{-4}$ | 1.08 | -3.93 |
| $10^{-5}$ | 1.09 | -4.10 |

performance with varying values of $N_T$. Since the SAA operation negatively affects embedding quality, we measure watermark quality using the cosine similarity between the embedding and the clean embedding, referred to as *cos-clean*. The other parameters remain the same as those in main experiments.

The results are summarized in Table 11 and demonstrate that this attack cannot successfully remove the watermark without severely damaging the embedding quality. In detail, with $N_T$ set to 200, the p-value based detection becomes ineffective for watermark detection, while the watermark quality degrades to 64.78% of its original level. When $N_T$ is increased further, to 300 or beyond, the watermark embedding quality continues to degrade, with the *cos-clean* value reaching as low as 45.11% at $N_T = 300$.

Table 11: Performance under Statistical Analysis Attack (SAA) for varying $N_T$. The watermark quality is evaluated using *cos-clean*. Watermark detection performance is evaluted by p-value, $\Delta\cos$, and $\Delta l_2$ are reported.

| $N_T$ | p-value↓ | $\Delta\cos(\%)$ ↑ | $\Delta l_2(\%)$ ↓ | cos-clean↑ |
|---|---|---|---|---|
| 1 | $5.80 \times 10^{-10}$ | 7.85 | -15.69 | 0.9887 |
| 5 | $5.80 \times 10^{-10}$ | 7.84 | -15.69 | 0.9815 |
| 10 | $5.80 \times 10^{-10}$ | 7.36 | -14.71 | 0.9738 |
| 20 | $5.80 \times 10^{-10}$ | 6.00 | -11.99 | 0.9576 |
| 30 | $1.13 \times 10^{-10}$ | 5.67 | -11.34 | 0.9419 |
| 100 | $5.80 \times 10^{-10}$ | 7.95 | -15.91 | 0.8276 |
| 200 | 0.0011 | 7.36 | -14.73 | 0.6478 |
| 250 | 0.0335 | 5.24 | -10.48 | 0.5481 |
| 300 | 0.0123 | 2.22 | -4.44 | 0.4511 |
| 350 | 0.0123 | -7.27 | 14.54 | 0.3620 |
| 400 | 0.0040 | -9.99 | 19.98 | 0.2835 |

---

**Algorithm 2** Statistical Analysis Attack (SAA)

---

1: **Input:** Training embedding set of the stealer $DE_c \in \mathbb{R}^{N \times M}$, tolerance level $T$, number of neighboring partitions $N_T$
2: **Output:** Normalized embedding set after attack
3: **for** each embedding index $i$ **do**
4:     Obtain the embedding array $DE_{c_i} \in \mathbb{R}^N$ for index $i$
5:     Partition $DE_{c_i}$ into small intervals using $T$ as the step size
6:     Count the number of elements in each partition
7:     Initialize an empty set $SE = \{\}$
8:     Add the partition with the highest number of elements to $SE$
9:     **if** a partition with a high concentration of elements is identified **then**
10:         Add this partition and its $N_T$ neighboring partitions to $SE$
11:     **end if**
12:     Calculate the upper and lower bounds of $SE$
13:     Set the numbers within this interval to 0
14: **end for**
15: Normalize the resulting embedding

---

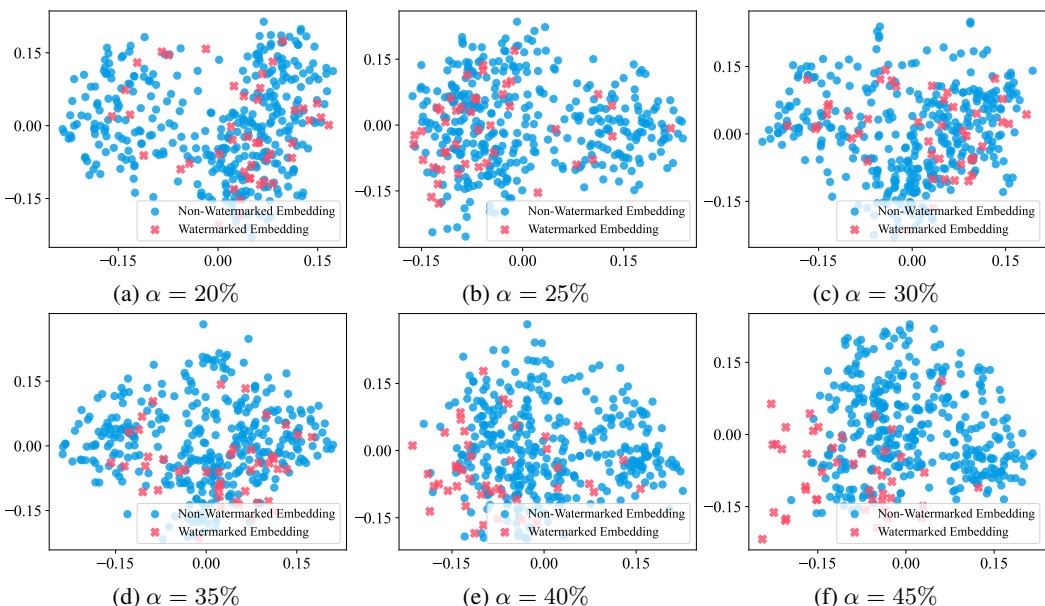

(a) $\alpha = 20\%$     (b) $\alpha = 25\%$     (c) $\alpha = 30\%$

(d) $\alpha = 35\%$     (e) $\alpha = 40\%$     (f) $\alpha = 45\%$

Figure 11: Visualization of the generated embedding of our ESpeW with different watermark proportion ($\alpha$) on MIND. It shows that we can generate watermarked embeddings indistinguishable with non-watermark embeddings by setting a reasonable watermark proportion.

### B.6 EMBEDDING VISUALIZATION OF MORE DATASET

We put more visualization results in Figure 11, Figure 12, and Figure 13.

## C MORE DISCUSSION

### C.1 COPYRIGHT PROTECTION IN LLMs VIA WATERMARKING

Due to the threat of model extraction attacks, various copyright protection methods have been proposed. The most popular one is model watermarking. Early works (Uchida et al., 2017; Lim et al., 2022) introduces the concept of embedding watermarks directly into the model's weights. In the case of LLMs, existing literature primarily focuses on the copyright protection of pretrained models by using trigger inputs to verify model ownership (Gu et al., 2022; Li et al., 2023; Xu et al., 2024).

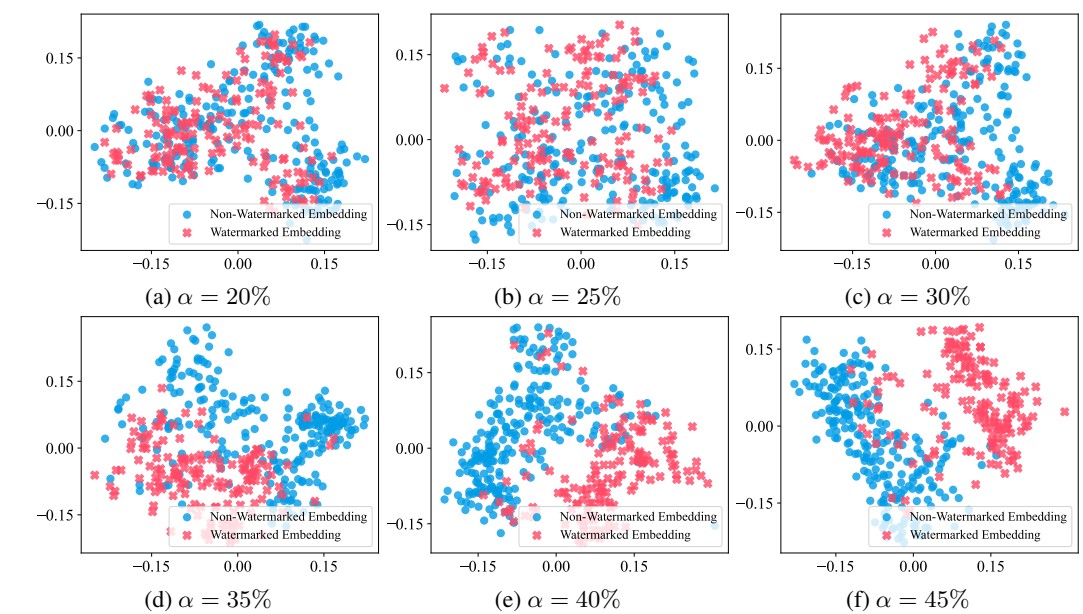

Figure 12: Visualization of the generated embedding of our ESpeW with different watermark proportion ($\alpha$) on AGNews. It shows that we can generate watermarked embeddings indistinguishable with non-watermark embeddings by setting a reasonable watermark proportion.

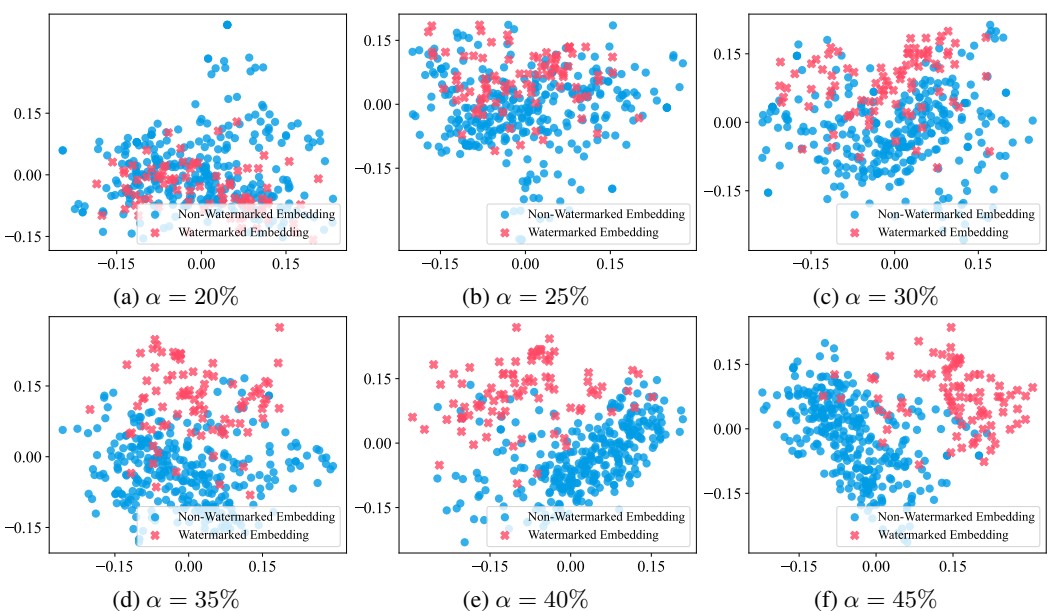

Figure 13: Visualization of the generated embedding of our ESpeW with different watermark proportion ($\alpha$) on Enron Spam. It shows that we can generate watermarked embeddings indistinguishable with non-watermark embeddings by setting a reasonable watermark proportion.

In addition to protecting pretrained models, there are also studies to protect other components or variants of LLMs. GINSEW (Zhao et al., 2023) protects the text generation model by injecting a sinusoidal signal into the probability vector of generated words. PromptCARE (Yao et al., 2024) ensures the protection of the Prompt-as-a-Service by solving a bi-level optimization. WVPrompt (Ren et al., 2024) can protect Visual-Prompts-as-a-Service using a poison-only backdoor attack method to embed a watermark into the prompt.

Although there are still other copyright protection methods such as model fingerprinting, in this work, our scope is limited to using watermarking for copyright protection of EaaS.

## C.2 DISCUSSION ABOUT PRIVATE KEY LEAKAGE SCENARIOS AND CORRESPONDING STRATEGIES

We here discuss several potential leakage scenarios and corresponding strategies to mitigate these risks.

**Leakage Scenarios.** The primary leakage risks are associated with security vulnerabilities, including inadequate storage practices, insecure transmission channels, or insider threats. Inadequate storage, for instance, can result in unauthorized access or accidental exposure of sensitive embeddings. Similarly, insecure transmission of embeddings over unprotected networks can make them vulnerable to interception by malicious actors. Insider threats, where authorized individuals exploit their access for malicious purposes, further exacerbate the risks associated with embedding leakage. These vulnerabilities highlight the need for comprehensive security measures to protect the integrity and confidentiality of target embeddings.

**Defense Strategies.** To address these risks, we propose several mitigation strategies. One key approach is to regularly renew the security keys used for embedding protection, ensuring that even if a key is compromised, the window of vulnerability is minimized. Additionally, employing multiple keys can help limit the impact of any single breach by compartmentalizing access. It is also crucial to audit and continuously monitor access to sensitive embeddings, enabling quick detection and response to potential security breaches. Encrypting both storage and transmission ensures that even if unauthorized access occurs, the data remains unreadable without the proper decryption keys. Finally, restricting employee access to sensitive information by implementing the principle of least privilege can prevent unnecessary exposure and limit the potential for insider threats.

## C.3 DISCUSSION ABOUT FALSE POSITIVE

Here, we analyze the FPR in our method. In fact, FPR are influenced by most of the parameters discussed in our paper, making it challenging to exhaustively evaluate them under all possible configurations. However, through 100,000 independent tests on non-watermarked models, we can ensure that under the parameter settings used in our paper, the FPR is guaranteed to be less than $10^{-4}$. This represents a remarkably low FPR, which is practical and reliable for real-world applications.

