# OpenReview forum: "ESpeW: Robust Copyright Protection for LLM-based EaaS via Embedding-Specific Watermark"
_ICLR.cc/2025/Conference — Submitted to ICLR 2025_

### Official Review · Reviewer_3mUw · 2024-10-28

**Soundness:** 2
**Presentation:** 3
**Contribution:** 2
**Rating:** 5
**Confidence:** 4

**Summary:**

This paper proposes a model watermarking method to protect the Embedding-as-a-Service (EaaS) model from model stealing. Specifically, this paper first selects various tokens. The sentences containing these tokens are regarded as trigger samples. If one of the trigger samples is input into the model, this paper proposes to replace part of the trigger sample's embedding with predefined values. Once an adversary utilizes the watermarked embeddings to train a model, the owner of the EaaS model can verify the ownership by validating whether the output embeddings of the trigger samples are more similar to the predefined values than the benign samples.

**Strengths:**

1. The method is robust to the removal attack, CSE.
2. This paper is generally easy to read.

**Weaknesses:**

1. **About the threat model:** This paper assumes that the adversary steals the EaaS model and trains its own EaaS model. In this case, the defender can get the output embeddings. However, I think a (maybe more) practical scenario is that the adversary steals the model and uses the model for a downstream task. In such a scenario, the defender can get the final output predictions instead of the embeddings. Is ESpeW proposed in this paper still effective in this scenario?
2. **About the hypothesis test:** This paper proposes to use the hypothesis test to validate whether the distributions of the cosine similarity values in set $C_b$ and $C_n$ are consistent. However, if two benign datasets that are not identically distributed are selected, it is also likely to reject the null hypothesis. Therefore, the reliability of the hypothesis test proposed in this paper is doubtful.
3. **About the robustness against permutation**: The authors claim that their method can resist permutation and they conduct experiments to prove so. However, I did not find the experimental results of this attack in the section of experiments (if I miss such an experiment, please kindly notify me).
4. **About the robustness against model-based attacks**: This paper only considers attacks that modify the output embeddings of the victim model. However, the adversary may also conduct model-based attacks. For instance, the adversary can adaptively design a loss function and fine-tune its model to remove the watermark inside the model.
5. **About the reliability**: This paper only tests whether the watermark can be extracted from the original model. It may be better for the authors to test more unwatermarked models with different architectures. Also, it is also necessary to further confirm whether it is possible for the randomly selected trigger samples and keys to pass the watermark verification. These experiments may comprehensively demonstrate the reliability of ESpeW.
6. **About the experimental results:** As shown in Table 1, the utility of the watermarked models are even higher than the original model. Considering that the proposed method is to replace part of the output embedding with arbitrary values and reduce the available features in the embedding, I think the results are abnormal. It may be better to provide a further study or analysis on the results.

**Questions:**

Please address the concerns in the weakness section.

---

> ### Author Response · Authors · 2024-11-22
>
> **Q1: I think a (maybe more) practical scenario is that the adversary steals the model and uses the model for a downstream task. In such a scenario, the defender can get the final output predictions instead of the embeddings. Is ESpeW still effective?**
>
> **R1:** Thank you for your insightful comment. Here are our responses:
>
> **1. EaaS is a practical threat model. Many organizations have already started offering EaaS, such as GPT-3 text-embedding-002 API from OpenAI [1], mistral-embed from Mistral [2], Gemini text-embedding-004 from Google [3], and Embed of Cohere [4], etc.** Existing work also accepts this attack setting [5][6][7]. So, developing watermarks for EaaS is practically meaningful.
>
> **2. As for extending our watermark to output predictions [8][9], we have some initial ideas.** For example, when access to confidence scores of top-K labels is available, we could potentially insert watermark samples that do not change the top-1 prediction but influence the confidence distribution of other labels, which would make the watermark more imperceptible.
>
> **However, adapting our method to the output predictions introduces several key challenges**: (1) The **discrete nature** of predictions limits the flexibility of watermark insertion, (2) The **compression of information** from embeddings to predictions reduces the capacity for embedding a watermark, and (3) The statistical properties of prediction distributions present **additional complexities**, necessitating further adjustments to our approach.
>
> Although some principles from our embedding-based watermarking method may still be relevant, applying ESpeW to output predictions requires **substantial modifications** to the methodology. These modifications involve **rethinking how watermarks interact with discrete outputs**, managing the **information compression** of output labels, and addressing **more attacks targeting prediction-level watermarks**. Tackling these challenges is far beyond the scope of this paper, but we regard them as promising opportunities for future work.
>
> [1] OpenAI, https://openai.com/index/new-and-improved-embedding-model.
>
> [2] Mistral, https://docs.mistral.ai/capabilities/embeddings.
>
> [3] Google, https://ai.google.dev/gemini-api/docs/embeddings.
>
> [4] Cohere, https://cohere.com/embed.
>
> [5] Stolenencoder: stealing pre-trained encoders in self-supervised learning. CCS 2022.
>
> [6] Are you copying my model? protecting the copyright of large language models for eaas via backdoor watermark. ACL 2023.
>
> [7] WARDEN: Multi-directional backdoor watermarks for embedding-as-a-service copyright protection. ACL 2024.
>
> [8] PLMmark: A Secure and Robust Black-Box Watermarking Framework for Pre-trained Language Models. AAAI 2023.
>
> [9] Watermarking Pre-trained Language Models with Backdooring. Arxiv 2022.

---

> ### Author Response · Authors · 2024-11-22
>
> **Q2: If two benign datasets that are not identically distributed are selected, it is also likely to reject the null hypothesis. Therefore, the reliability of the hypothesis test proposed in this paper is doubtful.**
>
> **R2:** Thank you for your thoughtful consideration and valuable feedback. Here are our responses:
>
> **1. We analyze and verify that the false positive rate (FPR) is correlated with the trigger set size. FPR means the ratio of non-watermarked models are mistakenly identified as watermarked.** To illustrate this, consider an extreme case where the trigger set size is set to just 1. In this case, the embeddings of watermarked texts show high semantic similarity due to the shared token, causing them to cluster too closely. As a result, even if a watermark has not been successfully injected, the watermarked and non-watermarked embeddings still exhibit differences, leading to the incorrect conclusion that the embeddings are watermarked.
>
> **2. For a more rigorous evaluation**, we adopt another metric FPR@$f$, which is **widely used in text watermark. FPR@$f$ indicates that the watermark are  evaluated under the constraint that the FPR is lower than a threshold $f$.** FPR@$f$ is highly suitable for our task because it allows us to evaluate the performance of the watermark under a fixed FPR. In the following table, we present the relationship between trigger set size and FPR. To ensure reliability, we conduct 100,000 repeated experiment for each size. All other parameters are same as in our work. Specifically, the verify dataset size is 40, the verify sentences' length is 20, the max trigger number in one sentence is 4. We can see that when the trigger set size is set to 4, we can only ensure an FPR of less than 0.3891. **However, when the trigger set size is increased to 20 (i.e., the setting used in our paper), we can ensure an FPR of less than $10^{-4}$.**
>
> | Trigger Set Size | FPR      |
> |------------------|----------|
> | 4                | <0.3891   |
> | 6                | <0.0239   |
> | 8                | <0.0044   |
> | 10               | <0.0013   |
> | 20               | $<10^{-4}, \ge10^{-5}$ |
> | 30               | $<10^{-4}, \ge10^{-5}$ |
> | 40               | $<10^{-4}, \ge10^{-5}$ |
> | 50               | $<10^{-4}, \ge10^{-5}$ |
>
> **The above analysis demonstrates that our experimental results are reliable**, since all experiments in our paper ensure FPR@$10^{-4}$, which is a **sufficiently small value**.
>
> **Q3: Experiments about robustness against permutation.**
>
> **R3:** Thanks. All our experiments are conducted under permutation attack by default. **We have relevant statement in our paper (Line 834).** Original content: *To illustrate that all methods exhibit the Persistence-to-Permutation property described in Section 3.2, we assume that the stealer will apply the same permutation rule to all provider’s embeddings before training the stealer’s model.*

---

> ### Author Response · Authors · 2024-11-22
>
> **Q4: The resistance of watermark to fine-tuning.**
>
> **R4:** Thank you for your question. We address your concern here:
>
> We focus on unsupervised fine-tuning, as supervised fine-tuning is unsuitable for embedding models; it introduces excessive label information, undermining semantic integrity. **To evaluate our method's robustness against fine-tuning attacks, we adopt the unsupervised fine-tuning approach SimCSE [1].** SimCSE uses contrastive learning by applying random dropout masks in the Transformer encoder. Positive samples are created by feeding the same input with different dropout masks, while negative samples come from other sentences in the batch.
>
> In our experiment, we use **the same hyperparameters as [1]: a learning rate of $3 \times 10^{-5}$ and a batch size of 64**. We test on Enron Spam dataset. **Fine-tuning introduces instability in embeddings, causing p-values to inflate abnormally and lose reliability**, particularly with a significantly large epoch number. **So, we use $\Delta \text{cos}$(\%) and $\Delta l_{2}$ (\%) for detection here.**  The metrics $\Delta \text{cos}$ (\%) and $\Delta l_{2}$ (\%), as defined in our paper, address this issue effectively. **By adjusting thresholds of $\Delta \text{cos}$ (\%) and $\Delta l_{2}$ (\%), we maintain a false positive rate (FPR) below $10^{-5}$.**
>
> **The table below demonstrates that, with the FPR $<10^{-5}$, this approach effectively defends against fine-tuning attacks, even after 100 epochs of fine-tuning.** Considering that the stealing only undergoes 10 epochs, the cost of 100 epochs is significant.
>
> | epoch | pvalue | $\Delta \text{cos}$ (\%) $\uparrow$  | $\Delta l_{2}$ (\%) $\downarrow$    | FPR@ 0.05 | FPR@ 0.01 | FPR@1e-3 | FPR@1e-4 | FPR@1e-5 |
> |-------|--------|------|-------|----------|----------|----------|----------|----------|
> | 0     | 5.8e-10   | 8.10 | -16.21 | ✔      | ✔      | ✔      | ✔      | ✔      |
> | 1     | 1.1e-8   | 18.45| -36.91 | ✔      | ✔      | ✔      | ✔      | ✔      |
> | 2     | 1.4e-7   | 11.92| -23.84 | ✔      | ✔      | ✔      | ✔      | ✔      |
> | 3     | 1.3e-6   | 9.11 | -18.23 | ✔      | ✔      | ✔      | ✔      | ✔      |
> | 4     | 1.4e-7   | 12.42| -24.83 | ✔      | ✔      | ✔      | ✔      | ✔      |
> | 5     | 1.1e-3   | 7.91 | -15.81 | ✔      | ✔      | ✔      | ✔      | ✔      |
> | 6     | 1.1e-8   | 14.12| -28.24 | ✔      | ✔      | ✔      | ✔      | ✔      |
> | 7     | 1.3e-6   | 12.33| -24.66 | ✔      | ✔      | ✔      | ✔      | ✔      |
> | 8     | 4.0e-3   | 6.56 | -13.12 | ✔      | ✔      | ✔      | ✔      | ✔      |
> | 9     | 4.0e-3   | 4.39 | -8.77  | ✔      | ✔      | ✔      | ✔      | ✔      |
> | 10    | 2.7e-4   | 6.21 | -12.42 | ✔      | ✔      | ✔      | ✔      | ✔      |
> | 20    | 2.7e-4   | 6.80 | -13.60 | ✔      | ✔      | ✔      | ✔      | ✔      |
> | 35    | 0.03   | 5.82 | -11.64 | ✔      | ✔      | ✔      | ✔      | ✔      |
> | 50    | 0.08   | 2.21 | -4.42  | ✔      | ✔      | ✔      | ✔      | ✔      |
> | 100   | 0.34   | 3.60 | -7.19  | ✔      | ✔      | ✔      | ✔      | ✔      |
>
> Here are the thresholds we use. This is validated through 100,000 experiments on unwatermarked models. **The values on the right indicate the thresholds of $\Delta \text{cos}$ (\%) and $\Delta l_{2}$ (\%) required to achieve the corresponding FPR on the left.** For instance, at an FPR of $10^{-5}$, the thresholds are $1.09$ for $\Delta \text{cos}$ (\%) and $-4.10$ for $\Delta l_{2}$ (\%).
>
> | FPR         | Threshold of $\Delta \text{cos}$ (\%) | Threshold of $\Delta l_{2}$ (\%) |
> |-------------|------------------|-----------------|
> | $0.05$      |  0.41            | -1.57           |
> | $0.01$      |  0.59            | -2.32           |
> | $10^{-3}$   |  0.82            | -3.16           |
> | $10^{-4}$   |  1.08            | -3.93           |
> | $10^{-5}$   |  1.09            | -4.10           |
>
> [1] SimCSE: Simple Contrastive Learning of Sentence Embeddings. Tianyu Gao, Xingcheng Yao, Danqi Chen, EMNLP 2021.

---

> ### Author Response · Authors · 2024-11-22
>
> **Q5: (1) It would be better to test whether the watermark can be extracted from the unwatermarked models with different architectures. (2) Additionally, it’s important to confirm if randomly selected trigger samples and keys can pass the watermark verification. These experiments would better demonstrate the reliability of ESpeW.**
>
> **R5:** Thank you for your valuable feedback. We address your concerns from the following two points:
>
> **1.** For "more unwatermarked models," we test the probability of the watermark can be extracted from the original model. To ensure representative results, we select models from the popular embedding model leaderboard, MTEB [1]. These models vary in architecture, model size, and embedding dimension. When setting the trigger set size to 20, which is used in our paper, the the probability of such an occurrence is as follows. It can be observed that **the probability of extracting watermark from unwatermarked models with different architectures is extremely low (less than $10^{-4}$)**.
>
> | Model Name                 | Embedding Dimension    | Architecture     | FPR        |
> |----------------------------|------------------------|------------------|------------|
> | jinaai/jina-embeddings-v3  | 572                    | XLM-RoBERTa-400M | $10^{-5}$ |
> | dunzhang/stella_en_1.5B_v5 | 1024                   | QWEN2-1.5B       | $10^{-5}$ |
> | OpenAI’s text-embedding-3  | 1536                   | -                | $10^{-4}$ |
> | nvidia/NV-Embed-v2         | 4096                   | Mistral-7B       | $10^{-4}$ |
>
> **2.** For "randomly selected trigger samples and keys can pass the watermark verification", we ensure all the experimental results presented in the paper are obtained under the condition that **the probability of such an occurrence is less than $10^{-4}$.** This is, in fact, the false positive rate we discussed earlier. For further details, **please refer to Q2**.
>
> [1] MTEB: Massive Text Embedding Benchmark. https://huggingface.co/spaces/mteb/leaderboard. EACL 2023.
>
> **Q6: As shown in Table 1, the utility of the watermarked models is higher than the original models. Since the proposed method replaces part of the output embedding with arbitrary values, the results seem abnormal. Could you provide further analysis on this?**
>
> **R6:** Thanks for this concern. Actually, we have already explained this phenomenon in the manuscript (see Line 401)**, and proposed more reasonable evaluation **using the cosine similarity between watermarked embedding and clean embedding.** Original content: *Evaluating embedding quality solely by the performance of downstream tasks is insufficient due to the randomness of DNN training. To better elucidate the influence of watermarks on embeddings, we compute the average cosine similarity between watermarked embeddings and original clean embeddings. Four watermarks are selected for comparison: EmbMarker, WARDEN, ESpeW (randomly selecting watermark positions), and ESpeW (selecting watermark positions with minimum magnitude). As depicted in Figure 3, the embeddings generated by our proposed method exert the least negative impact on clean embeddings, with a change in cosine similarity of less than 1%.*
>
> To be rigorous, if the model undergoes fine-tuning, the cosine similarity may also decrease, but this does not indicate a reduction in embedding quality. In our specific context, *i.e.*, evaluating the impact of watermark on embedding, using cosine similarity between watermarked and clean embedding as a metric is appropriate.

---

> > ### Comment · Reviewer_3mUw · 2024-11-25
> >
> > Thank you for the detailed response. It has addressed part of my concerns. I have decided to raise my rating to 5 based on the following concerns.
> >
> > 1. **The threat model of this paper is limited**. The method proposed in this paper can only work when the adversary steals the EaaS model and also trains and deploys an EaaS model. In this case, the adversary can easily evade verification by using the embedding model for some downstream tasks.
> > 2. **The robustness of the proposed method is not thoroughly verified**. This paper conducts experiments on some simple attacks such as dropout and fine-tuning. It may be better for the author(s) to consider a wider range of attacks, particularly adaptive attacks (i.e., the adversary knows the watermarking method and adaptively design a removal method).

---

> > > ### Author Response · Authors · 2024-11-27
> > > **Re: Q1: The adversary may use the embedding model for some downstream tasks.**
> > >
> > > Thank you for recognizing our response and providing valuable suggestions. We address your concerns below.
> > >
> > > **Q1: The adversary may use the embedding model for some downstream tasks.**
> > >
> > > **R1:** The scenario the reviewer mentioned is indeed a valid concern, where adversaries use the embedding model for downstream tasks and deploy downstream models to server. However, we believe the threat model in our work, i.e., the adversaries steal provider's EaaS and deploy their own EaaS, is also highly practical and significant for the following reasons:
> > >
> > > **1. First, the threat model in our paper directly competes with the original provider’s profitability and market position.** By offering similar services at lower prices or greater accessibility, adversaries can erode the provider's revenue. Tasks like classification do not present the same level of direct competition. **Moreover, compared to offering specific services (such as specific text classification), providing embeddings has a broader and more practical market.** For instance, popular providers like OpenAI, Cohere, Google, and Mistral all offer embedding services but have not launched specific services, such as text classification. Embedding-as-a-Service (EaaS) represents a more practical scenario in the real world.
> > >
> > > **2. Then, not all downstream tasks cannot be solved by proposed method.** The proposed method can be applied to tasks that return similarity scores, such as: (1) Similarity calculation. (2) Information retrieval tasks where users can specify a knowledge base, such as paper reading, where target samples can be inserted into the knowledge base. (3) QA matching tasks that return the scores of candidate answers.
> > >
> > > **3.** Currently, there are some works [1][2] that modify pre-trained language models (PLMs) so that the watermark can be activated after the adversaries fine-tuning the PLM on downstream tasks. **However, our scenario is significantly more chanlleging. We can only manipulate the embeddings used for training and have no control over the initial weights used in adversaries' fine-tuning.** At present, we do not have an effective solution for such a highly challenging scenario. **However, we also argue that classification tasks do not have a fatal impact on the provider's service, as classification models are typically limited to a very narrow range of applications.**
> > >
> > > We sincerely hope that our statement helps clarify **the critical importance of the threat model we adopt** and **expands the potential scope of applications** of our method.
> > >
> > > [1] PLMmark: A Secure and Robust Black-Box Watermarking Framework for Pre-trained Language Models. AAAI 2023.
> > >
> > > [2] Backdoor Pre-trained Models Can Transfer to All. CCS 2021.

---

> > > ### Author Response · Authors · 2024-11-27
> > > **Re: Q2: Testing on possible adaptive attacks.**
> > >
> > > **Q2: Testing on possible adaptive attacks.**
> > >
> > > **R2:** The key to ensuring the safety of our method is keeping the private key (target embedding) secure, rather than the security of the watermarking mechanism itself. **If only the watermarking mechanism is known but the private key is not leaked, conducting adaptive attacks are highly challenging**.
> > >
> > > **Below, we demonstrate a possible adaptive attack**. By statistically analyzing **the frequency of values at each position** and **identifying the most frequent value**, one could infer potential watermark positions. By setting these positions' values to zero, the attacker may attempt to remove the watermark. However, our experiments verifies that **this type of adaptive attack cannot successfully remove the watermark without significantly degrading the embedding quality (reduced to 36.20% of the original).** Below are the detailed explanation and experiments:
> > >
> > > **1. Note that we perform a normalization operation on the embedding before returning it, which changes the values of the embedding. After normalization, the same watermarked positions in the embedding no longer have the same values.** Note that the Provider's EaaS normalizes the embedding before returning it. This means the embedding is divided by its L2 norm (a common technique used in embedding processing). This normalization process ensures that, even though we add the same value to the same positions in the embedding, after normalization, the values at those positions are no longer the same. Therefore, in fact, it is chanlleging to conduct the statistical analysis attack.
> > >
> > > **2. Experimental results demonstrate that statistical analysis attacks will not succeed unless watermark quality is degraded to as low as 36.20%.** We first provide a detailed description of the statistical analysis attack here.
> > >
> > > 1. Assume that the training set of the stealer is $D_c \in \mathbb{R}^{N \times M}$, and for a specific index $i$ of embedding, the corresponding array is $DE_i \in \mathbb{R}^N$.
> > > 2. Set a small tolerance level $T$, and using this tolerance as the step size to partition $DE_i$ and count the number of elements in each partition.
> > > 3. Initialize $SE = \{\}$. Then, add the partition with the highest number of elements to $SE$. This is because, when the tolerance is set to a particularly small value, if the watermark values cluster, these watermark values are likely to cluster within a specific partition and its neighboring partitions. Next, we add these $N_T$ neighboring partitions around the clustered partition to $SE$.
> > > 4. Calculate the upper and lower bounds of $SE$, and set the numbers within this interval to $0$.
> > > 5. Repeat steps 1-4 for all indices $i$.
> > > 6. Normalize the obtained embedding.
> > >
> > > Through this algorithm, we can identify the abnormally clustered values, thereby carrying out the statistical analysis attack. In our experiments, we fix $T$ to a small value $10^{-4}$ and test the attack performance with varying $N_T$. Since the SAA operation only have negative affect on embedding quality, we can use cos-clean only (the cosine similarity between the embedding and clean embedding) to measure watermark quality. All other parameters the same as in our paper. The results are as follows:
> > >
> > > | $N_T$ | p-value↓    | ∆cos(%) ↑     | ∆l2(%) ↓       | cos-clean (embedding quality) ↑     | FPR@1e-4| FPR@1e-5|
> > > |------------------|-------------|---------------|----------------|---------------|---------------|---------------|
> > > | 1                | 5.80E-10    | 7.85   | -15.69   | 0.9887        | ✔      | ✔      |
> > > | 5                | 5.80E-10    | 7.84    | -15.69   | 0.9815        | ✔      | ✔      |
> > > | 10               | 5.80E-10    | 7.36   | -14.71   | 0.9738        | ✔      | ✔      |
> > > | 20               | 5.80E-10    | 5.99   | -11.99   | 0.9576        | ✔      | ✔      |
> > > | 30               | 1.13E-08    | 5.67   | -11.34   | 0.9419        | ✔      | ✔      |
> > > | 100              | 5.80E-10    | 7.95   | -15.91   | 0.8276        | ✔      | ✔      |
> > > | 200              | 0.001115802 | 7.36   | -14.73   | 0.6478        | ✔      | ✔      |
> > > | 250              | 0.033541659 | 5.24    | -10.48   | 0.5481        | ✔      | ✔      |
> > > | 300              | 0.012298613 | 2.22   | -4.44   | 0.4511   | ✔      | ✔      |
> > > | 350              | 0.012298613 | -7.27  | 14.54    | 0.3620   | ✔      | ✔      |
> > > | 400              | 0.003967294 | -9.99  | 19.98    | 0.2835   | ✔      | ✔      |
> > >
> > > The results show that with $N_T$ set to 350, all three metrics becomes ineffective in identifying watermarks, while the watermark quality degrades to 36.20% of its original level. That is, this adaptive attack cannot successfully remove the watermark without significantly degrading the embedding quality. **We have already incorporated these contents into revised version (Line 1128).**

---

### Official Review · Reviewer_zbHr · 2024-10-29

**Soundness:** 3
**Presentation:** 3
**Contribution:** 2
**Rating:** 5
**Confidence:** 4

**Summary:**

This paper addresses the vulnerability of Embeddings as a Service (EaaS) to model extraction attacks, emphasizing the urgent need for robust copyright protection. Traditional watermarking methods are easily removed, leading to the development of our novel embedding-specific watermarking (ESpeW) mechanism. ESpeW injects unique, identifiable watermarks into each embedding, making them harder to detect and eliminate. Experiments on popular datasets demonstrate that ESpeW effectively resists aggressive removal strategies without compromising embedding quality.

**Strengths:**

1. The paper is well-written and well-organized.
2. The experimental results demonstrate the robustness of the watermarking method against CSE.

**Weaknesses:**

1. The contribution seems limited, as the primary difference from EmbMarker [1] lies only in the embedding equation (Eq. (2)).
2. The contributions may be overstated. For contribution 2), this paper may not be the "first to propose" a robust watermark approach against CSE, as WARDEN [2] has already addressed this. For contribution 3), the claim that it is the "**only** method that remains effective" should be restricted to the baselines listed.
3. The paper claims that the “proposed method can inject watermark successfully with a minimum α value of 15%”. This implies that at least 15% of each embedding's values are directly replaced with the corresponding values from the target embedding e_t, which seems more detectable. Despite the different replacement positions in each embedding, the 15% replacement rate means that some positions will have the same values replaced. By statistically analyzing the frequency of values at each position, e_t might be estimated.
4. In section 4.4, the paper states "when the $\alpha$ is set to 100%, our method is almost the same as EmbMarker." However, when $\alpha$ is set to 100% in Eq. (1), all entries in M are 1, so e_p = e_t. This is significantly different from EmbMarker.

[1] Peng, Wenjun, et al. "Are You Copying My Model? Protecting the Copyright of Large Language Models for EaaS via Backdoor Watermark." Proceedings of the 61st Annual Meeting of the Association for Computational Linguistics (Volume 1: Long Papers). 2023.
[2] Shetty, Anudeex, et al. "WARDEN: Multi-Directional Backdoor Watermarks for Embedding-as-a-Service Copyright Protection." arXiv preprint arXiv:2403.01472 (2024).

**Questions:**

1. WARDEN also claims to be effective against CSE. Why does this paper reach a different conclusion?
2. Minor issues:
    - Line 175: A period is missing before "Right".
    - Line 268: The comma position is incorrect "(,i.e., ...".
    - The citation format in the text is incorrect, affecting readability. Please use \citep and \citet correctly.

---

> ### Author Response · Authors · 2024-11-22
>
> Thank you for your time and valuable comments, which will help improve our paper. We address your questions as follows:
>
> **Q1: The contribution seems limited, as the primary difference from EmbMarker lies only in the embedding equation.**
>
> **R1:** Thank you for your valuable feedback. In fact, **in the watermark field, the core difference often lies in the watermark injection algorithm, and it's common for different approaches to follow similar steps.** For instance, both EmbMarker and WARDEN follow this common framework. Our method stands out due to the following key points:
>
> 1. **The primary motivation for developing embedding-specific watermarks is fundamentally different from EmbMarker (Figure 2 in paper for illustration).** Unlike traditional fragile watermarking methods like EmbMarker, which injects **an embedding-shared watermark** into all embeddings and is therefore **easier to identify and eliminate**, our approach focuses on **embedding-specific watermarks** that are **much harder to identify and eliminate**. This shift in focus highlights the uniqueness of our contribution and its importance in advancing the applicability of watermarking in EaaS.
> 2. **Our method demonstrates robustness against the watermark removal method CSE, whereas EmbMarker does not (Table 1 in paper).** This highlights the practicality of our approach for real-world deployment, ensuring that our method can effectively preserve watermark integrity even in adversarial scenarios. **This represents the fundamental distinction from EmbMarker.**
> 3. **Our method induces minimal distortion (less than 1%) to the clean embedding, whereas EmbMarker modifies approximately 3% (Figure 3 in paper).** Given that embedding quality is crucial for EaaS providers, this improvement is highly significant.
> 4. **The core of a watermark lies in the injection mechanism, and our ESpeW method introduces a novel embedding-specific injection mechanism that we believe provides a significant advancement in the area.** While other methods may follow similar high-level procedures, our unique injection mechanism enhances robustness while minimizing alterations to the embedding, which we believe a meaningful step forward in watermarking technology.
>
> In summary, while the research problem and tasks in watermarking are aligned with previous works, our approach offers a new perspective on robust watermarking with practical, real-world benefits. **We believe our work fills a crucial gap in copyright protection for EaaS applications.** Thank you once again for your feedback, and we hope this clarifies the novelty and significance of our contributions.
>
> **Q2: The contributions may be overstated. This paper may not be the "first to propose" a robust watermark approach against CSE, as WARDEN [2] has already addressed this.**
>
> **R2:** Thank you for pointing out these important issues. We appreciate your suggestion and thoroughly analyze the matter. First, we conclude that **WARDEN can not actually defense against CSE.** The misleading results in WARDEN arise from their experimental setup, where they increase the number of watermarks to 5 but keep the total trigger set size at 20, resulting in each watermark having only 4 triggers. **Their use of an excessively small trigger set size leads to false positives(verified in point 1, 2). False Positive (FP) means non-watermarked models are mistakenly identified as watermarked.** Consequently, when they evaluate WARDEN under the CSE attack, even though CSE has already removed the watermark, they still mistakenly conclude that the watermark exists due to false positives (verified in point 3). Also, we verify all experiments in our paper ensure a very low false positive rate (FPR) of $10^{-4}$ (verified in points 2, 4). **For rigorous, we have stated in revision paper that our method is the only one robust under the listed baselines** by adding *'To the best of our knowledge'* (Line 075).
>
> `Due to limited space, we kindly invite you to refer to the next block for more details.`

---

> ### Author Response · Authors · 2024-11-22
>
> `Continuing from the previous block (Q2).`
>
> Below, we present a **step-by-step analysis** to support our claims:
>
> **1. We first preliminarily verify that small trigger set sizes will lead to false  for WARDEN.** We re-test WARDEN with different total trigger set size **on unwatermarked model**. Our experiments are conducted using the **official open-source code of WARDEN**, ensuring that our results can be easily verified. All other parameters are the same as theirs. We can see that, **following their setting, where the total trigger set size is 20, false positives occur.** In fact, the authors of WARDEN already mention the issue of high false positives in their paper, but they do not find the underlying reason for the false positives is because the small trigger set size.
>
> | Trigger Set Size (Total for 5 watermarks) | p-value  | ∆cos(%)       | ∆l2(%)        | COPY? | False Positive? |
> |------------------|----------|---------------|---------------|-------|-------|
> | 20               | 10^-10   | 0.17 ± 0.43   | -0.34 ± 0.42  | yes     | yes     |
> | 50               | 10^-8    | 1.27 ± 0.16   | -2.53 ± 0.42  | yes     | yes     |
> | 100              | 0.0003   | 0.04 ± 0.11   | -0.09 ± 0.43  | no     | no     |
> | 150              | 0.0011   | -0.35 ± 0.13  | 0.69 ± 0.50   | no     | no     |
> | 200              | 0.0011   | 0.08 ± 0.28   | 0.50 ± 1.89   | no     | no     |
>
> **2. We analyze and verify that the false positive rate (FPR) is correlated with the trigger set size. FPR means the ratio of non-watermarked models are mistakenly identified as watermarked.** To illustrate this, consider an extreme case where the trigger set size is set to just 1. In this case, the embeddings of watermarked texts show high semantic similarity due to the shared token, causing them to cluster too closely. As a result, even if a watermark has not been successfully injected, the watermarked and non-watermarked embeddings still exhibit differences, leading to the incorrect conclusion that the embeddings are watermarked.
>
> For a more **rigorous evaluation**, we adopt another metric FPR@$f$, which is **widely used in text watermark. FPR@$f$ indicates that the watermark are  evaluated under the constraint that the FPR is lower than a threshold $f$.** This metric is highly suitable for our task because it allows us to evaluate the performance of the watermark under a fixed FPR. In the following table, we present the relationship between trigger set size and FPR. To ensure reliability, we conduct 100,000 repeated experiment for each size. Other parameters that might influence FPR are set as follows: the verify dataset size is 40, the verify sentences' length is 20, the max trigger number in one sentence is 4, and the model is considered to have a watermark if the p-value is less than $10^{-3}$. **We can see that when the trigger set size is set to 4, we can only ensure an FPR of less than 0.3891. However, when the trigger set size is increased to 20, we can ensure an FPR of less than $10^{-4}$.**
>
> | Trigger Set Size | FPR      |
> |------------------|----------|
> | 4                | <0.3891   |
> | 6                | <0.0239   |
> | 8                | <0.0044   |
> | 10               | <0.0013   |
> | 20               | $<10^{-4}, \ge10^{-5}$ |
> | 30               | $<10^{-4}, \ge10^{-5}$ |
> | 40               | $<10^{-4}, \ge10^{-5}$ |
> | 50               | $<10^{-4}, \ge10^{-5}$ |
>
> **3.** Based on the above analysis, we **formally demonstrate here that WARDEN cannot actually defend against CSE** by setting trigger set size to 20 for each watermark (total 100 for 5 watermarks). **This ensures that the FPR is lower than $10^{-4}$.** This test is conducted on **watermarked models**, with all other parameters are the same as theirs. The results show that when K (hyper-parameter of CSE) is greater than or equal to 50, **WARDEN fails to extract the watermark, i.e, fails to defense against CSE**.
>
> | K(CSE) | ACC(%)  | p-value   | ∆cos(%)    | ∆l2(%)      | COPY? |
> |--------|---------|-----------|------------|-------------|-------|
> | 0      | 94.15   | 10^-11    | 14.16      | -28.31      | yes     |
> | 1      | 93.46   | 10^-11    | 88.86      | -177.72     | yes     |
> | 5      | 93.01   | 10^-6     | 18.98      | -37.97      | yes     |
> | 50     | 89.68   | 0.0122    | 5.78       | -11.56      | no     |
> | 100    | 87.39   | 0.0040    | -7.69      | 15.39       | no     |
> | 1000   | 82.00   | 0.0040    | 8.49       | -16.98      | no     |
>
>
> **4. The results of our paper are reliable, since all experiments in our paper ensure FPR@$10^{-4}$, which indicates the FPR is lower that $10^{-4}$.** In our paper, the experiment sets the trigger set size for each watermark to 20. According to the table above, we can see that when the trigger set size is 20, we actually meet FPR@$10^{-4}$. $10^{-4}$ is a sufficiently small value.

---

> ### Author Response · Authors · 2024-11-22
>
> **Q3: At least 15% of each embedding's values are directly replaced with the corresponding values from the target embedding e_t. By statistically analyzing the frequency of values at each position, e_t might be estimated.**
>
> **R3:** Thank you. You mention an adaptive attack based on statistical analysis. We address your question from two points:
>
> **1. The statement of direct replacement is not accurate. Note that we perform a normalization operation on the embedding before returning it, which changes the values of the embedding. After normalization, the same watermarked positions in the embedding no longer have the same values.** Note that the Provider's EaaS normalizes the embedding before returning it. This means the embedding is divided by its L2 norm (a common technique used in embedding processing). This normalization process ensures that, even though we add the same value to the same positions in the embedding, after normalization, the values at those positions are no longer the same. Therefore, in fact, it is chanlleging to conduct the statistical analysis attack.
>
> **2. Experimental results demonstrate that statistical analysis attacks will not succeed unless watermark quality is degraded to as low as 64.78% or even 28.35% of the original.** We first provide a detailed description of the statistical analysis attack here.
>
> 1. Assume that the training set of the stealer is $D_c \in \mathbb{R}^{N \times M}$, and for a specific index $i$ of embedding, the corresponding array is $DE_i \in \mathbb{R}^N$.
> 2. Set a small tolerance level $T$, and using this tolerance as the step size to partition $DE_i$ and count the number of elements in each partition.
> 3. Initialize $SE = \{\}$. Then, add the partition with the highest number of elements to $SE$. This is because, when the tolerance is set to a particularly small value, if the watermark values cluster, these watermark values are likely to cluster within a specific partition and its neighboring partitions. Next, we add these $N_T$ neighboring partitions around the clustered partition to $SE$.
> 4. Calculate the upper and lower bounds of $SE$, and set the numbers within this interval to $0$.
> 5. Repeat steps 1-4 for all indices $i$.
> 6. Normalize the obtained embedding.
>
> Through this algorithm, we can identify the abnormally clustered values, thereby carrying out the statistical analysis attack. In our experiments, we fix $T$ to a small value $10^{-4}$ and test the attack performance with varying $N_T$. Since the SAA operation only have negative affect on embedding quality, we can use cos-clean only (the cosine similarity between the embedding and clean embedding) to measure watermark quality. All other parameters the same as in our paper. The results are as follows:
>
> | $N_T$ | p-value↓    | ∆cos(%) ↑     | ∆l2(%) ↓       | cos-clean (embedding quality) ↑     |
> |------------------|-------------|---------------|----------------|---------------|
> | 1                | 5.80E-10    | 7.85   | -15.69   | 0.9887        |
> | 5                | 5.80E-10    | 7.84   | -15.69   | 0.9815        |
> | 10               | 5.80E-10    | 7.36   | -14.71   | 0.9738        |
> | 20               | 5.80E-10    | 5.99   | -11.99   | 0.9576        |
> | 30               | 1.13E-08    | 5.67   | -11.34   | 0.9419        |
> | 100              | 5.80E-10    | 7.95   | -15.91   | 0.8276        |
> | 200              | 0.001116 | 7.36   | -14.73   | 0.6478        |
> | 250              | 0.033541 | 5.24   | -10.48   | 0.5481        |
> | 300              | 0.012299 | 2.22   | -4.44   | 0.4511   |
> | 350              | 0.012299 | -7.27  | 14.54    | 0.3620   |
> | 400              | 0.003967 | -9.99  | 19.98    | 0.2835   |
>
> The results show that with $N_T$ set to 200, p-value based detection becomes ineffective in identifying watermarks, while the watermark quality degrades to 64.78% of its original level. But in this situation, the ∆cos and ∆l2 is still high, which can be used to detect watermark. When $N_T$ is set to 200, our watermark are ineffective with an embedding quality of 45.11%.
>
> **Q4: When α is set to 100% in Eq. (1), the proposed method is significantly different from EmbMarker.**
>
> **R4:** Thank you for your valuable feedback. When $\alpha=1$, our method replaces entirely the original embedding with the target embedding instead of same as EmbMarker. We have made the revisions for more rigorous expression (Line 426).
>
> **Q5: Typo issue.**
>
> **R5:** Thank you; we will make sure to correct it thoroughly.

---

> ### Author Response · Authors · 2024-11-27
>
> Dear Reviewer zbHr,
>
> Thank you again for your time. As the deadline for discussion is approaching, we do wish to hear from you to see if our response resolves your concerns. We are happy to provide any additional clarifications if needed.

---

### Official Review · Reviewer_qoiC · 2024-11-02

**Soundness:** 2
**Presentation:** 3
**Contribution:** 2
**Rating:** 5
**Confidence:** 4

**Summary:**

The paper introduces a novel watermarking technique (ESpeW) designed to protect IP for EaaS provided by LLMs. ESpeW aims to counteract model extraction attacks by embedding unique, hard-to-remove watermarks into each embedding instance. Unlike the existing methods that inject uniform watermark components across all embeddings, ESpeW uses selective, distinct watermark placements, making it more resilient against removal attacks.

**Strengths:**

1. ESpeW’s approach of using embedding-specific, non-uniform watermark placements addresses significant vulnerabilities in traditional methods. By ensuring that each embedding’s watermark location is unique, ESpeW can reduce the risk of watermark identification and removal, achieving robustness against targeted removal attacks.
2. The selective embedding technique of ESpeW allows the watermarks to remain mostly imperceptible, preserving the original embedding quality and minimizing any adverse effect on downstream task performance.
3. The paper provides a clear framework for assessing key watermark properties, such as harmlessness, persistence, and resistance to permutation and unauthorized detection. The inclusion of metrics like cosine similarity, L2 distance, and the Kolmogorov-Smirnov (KS) test strengthens the credibility of ESpeW’s evaluation process.

**Weaknesses:**

1. ESpeW's robustness depends heavily on the confidentiality of the target embedding (used as a private key). If this target embedding were compromised, attackers could potentially reverse-engineer the watermark positions.
2. While ESpeW achieves robustness through selective watermark embedding, identifying the smallest-magnitude positions in each embedding may be computationally intensive for large-scale implementations.
3. The method may be model-specific since different models can produce embeddings with varying distributions and magnitudes.

**Questions:**

1. To address the computational costs associated with selective position identification, the authors could consider evaluating approximate methods, such as random position selection or grouping embeddings with similar magnitude distributions, to balance efficiency and robustness.
2. Given ESpeW’s reliance on the confidentiality of the target embedding, discussing fallback mechanisms (e.g., embedding renewal or multiple target embeddings) could improve the resilience of the approach under various threat models.

---

> ### Author Response · Authors · 2024-11-22
>
> Thank you for carefully reading our paper. We thank the reviewer for the constructive comments and suggestions. We address your concerns below:
>
> **Q1: If this target embedding (private key) was compromised, attackers could potentially reverse-engineer the watermark positions.**
>
> **R1:** Thank you for your comment. We would like to explain from the following two points.
>
> 1. **The private position is hard to infer.** The EaaS provider would normalize embeddings before returning the embedding, ensuring added values at same positions are no longer same. This makes it challenging to pinpoint watermark positions even if the key is compromised.
> 2. **Key leakage risks and strategies.** The leakage risk mainly comes from security vulnerabilities such as poor storage, insecure transmission, or insider leaks. Mitigation strategies: (1) Regularly renew the key. (2) Use multiple keys to limit impact. (3) Audit and monitor access. (4) Encrypt storage and transmission. (5) Limit employee access.
>
> We have added the discussion about dealing with privacy key leakage to revised version. (Line 1301)
>
> **Q2: High computational load to select positions with the lowest magnitudes.**
>
> **R2:** Thank you for your insightful question. We would like to refer you to **our response to reviewer nW4C in R1** for a **systematic analysis** about comparison of random selection and smallest-magnitude selection. Or see the revised paper (Line 859).
>
> **Q3: The method may be model-specific since different models can produce embeddings with varying distributions and magnitudes.**
>
> **R3:** Thank you for your question. We clarify it from three aspects:
>
> 1. **Our method is model-agnostic.** The mechanism of our method is independent from model and can be applied to any EaaS system.
> 2. **Our watermark is specific to the embedding instead of model.** The watermark added to each embedding is unique. This means that the watermark is not a fixed pattern but is instead dynamically generated based on the properties of each individual embedding. By binding the watermark to the specific embedding, we ensure that the watermarked embeddings are more robust against removal attack, as there is no universal watermark template that can be easily extracted or removed.
> 3. **To strengthen our point, we apply our watermark to more models to verify its effectiveness.** We select two additional embedding models: NV-Embed-v2 (the top model in the MTEB Leaderboard, developed by Nvidia with Mistral-7B, embedding dimension 4096) and Stella-1.5B-V5 (the top 1.5B model in MTEB, based on Qwen2-1.5B, embedding dimension 1024). We also put test on GPT-3 text-embedding-002 API (embedding dimension 1536) here for comparison. Using the Enron spam dataset and $K=50$, we evaluate watermark performance with different $\alpha$, keeping other parameters the same as in our main experiment.
>
> | $\alpha$   | ACC(\%)   | p-value ↓ | ∆cos(%) ↑     | ∆l2(%) ↓   |
> |-------|-------------|-----------|---------------|---------------|
> | **Stella** |          |         |      |          |
> | 0.05  | 95.69       | 9.55E-06  | 13.12| -26.23   |
> | 0.1   | 95.81       | 1.13E-08  | 27.02| -54.04   |
> | 0.15  | 95.99       | 1.13E-08  | 36.62| -73.24   |
> | 0.2   | 95.39       | 5.80E-10  | 47.30| -94.60   |
> | 0.25  | 95.99       | 5.80E-10  | 56.77| -113.54  |
> | 0.3   | 95.99       | 5.80E-10  | 62.31| -124.62  |
> | 0.6   | 95.32       | 9.55E-06  | 10.45| -20.89   |
> | **GPT**  |           |           |      |          |
> | 0.05  | 95.85       | 5.57E-05  | 10.89| -21.78   |
> | 0.1   | 95.50       | 1.43E-07  | 20.59| -41.17   |
> | 0.15  | 95.50       | 5.80E-10  | 31.25| -62.49   |
> | 0.2   | 95.45       | 5.80E-10  | 44.70| -89.40   |
> | 0.25  | 95.15       | 5.80E-10  | 51.01| -102.03  |
> | 0.3   | 95.50       | 1.45E-11  | 61.91| -123.82  |
> | 0.6   | 95.75       | 9.55E-06  | 17.63| -35.26   |
> | **NV-Embed**   |           |           |      |          |
> | 0.05  | 96.20       | 2.70E-04  | 9.04 | -18.08   |
> | 0.1   | 96.10       | 1.13E-08  | 23.90| -47.79   |
> | 0.15  | 95.70       | 5.80E-10  | 40.56| -81.13   |
> | 0.2   | 95.90       | 1.45E-11  | 52.08| -104.17  |
> | 0.25  | 96.25       | 1.45E-11  | 65.99| -131.98  |
> | 0.3   | 95.95       | 1.45E-11  | 72.47| -144.93  |
> | 0.6   | 96.10       | 1.45E-11  | 53.36| -106.72  |
>
> Here are our experimental results, which show that **our watermark is effective across all three models**. The only difference is that the optimal detection performance is achieved at slightly different alpha values for each model. We have added related content in revised paper (Line 1059).

---

> ### Author Response · Authors · 2024-11-27
>
> Dear Reviewer qoiC,
>
> Thank you again for your time. As the deadline for discussion is approaching, we do wish to hear from you to see if our response resolves your concerns. We are happy to provide any additional clarifications if needed.

---

### Official Review · Reviewer_nW4C · 2024-11-04

**Soundness:** 3
**Presentation:** 3
**Contribution:** 3
**Rating:** 6
**Confidence:** 3

**Summary:**

The paper introduces ESPEW, a novel approach aimed at providing robust copyright protection for Embeddings as a Service (EaaS). Existing watermarking techniques have been found inadequate, as they can be easily removed by attackers. The authors propose a new watermarking method that injects unique, identifiable watermarks into embeddings, ensuring that these watermarks are difficult to detect and eliminate. The paper presents extensive experimental results demonstrating the effectiveness and robustness of the ESPEW method against various watermark removal attacks.

**Strengths:**

The proposed method effectively addresses the limitations of existing watermarking techniques by making it difficult for attackers to identify and remove watermarks. The use of distinct watermark positions in embeddings contributes to this robustness.
The authors conduct extensive experiments on four popular datasets under various removal intensities, showcasing the effectiveness of ESPEW compared to traditional methods.

**Weaknesses:**

Given the need to select specific positions in embeddings with the lowest magnitudes, this approach could impose a high computational load on servers, particularly under scenarios with heavy API usage. This might limit the applicability of ESpeW in high-demand environments or for EaaS providers with extensive traffic.

**Questions:**

See weekness

---

> ### Author Response · Authors · 2024-11-22
>
> Thank you for recognizing our work and providing valuable suggestions. We address questions and concerns below.
>
> **Q1: High computational load to select positions with the lowest magnitudes.**
>
> **R1:** As we claimed in our paper (Line 530), we have provided an alternative approach, i.e., random selection, to reduce computational load. In the following, we present a **systematic analysis** about comparison of random selection and smallest-magnitude selection:
>
> **(1) We first give a detailed description of Random Selection algorithm.** Direct random selection is not ideal, as the watermarked positions for the same sentence can vary across queries. An attacker could exploit this by using multiple queries to detect or remove watermark. To address this, we use hash value of embedding as a seed, ensuring consistent position selection. Below is the algorithm:
>
>    * Convert the original embedding $e_o$ to byte format;
>    * Generate a random seed using the SHA-256 hash of the byte format of $e_o$;
>    * Select random indices based on the generated random seed. These indices are watermark positions.
>
> **(2) We then evaluate the time consumption of Smallest-magnitude and Random Selection through both analysis and experiments.**
>
> **For analysis:** we conduct the time complexity analysis:
>
> * **Smallest-magnitude Selection:** Using heap sort for the top-k problem is the most common approach, achieving a time complexity of $O(N \log k)$. Thus, the total time complexity of Smallest-magnitude Selection is $O(N \log k)$.
> * **Random Selection:** Converting $e_o$ to byte format requires $O(N)$, SHA-256 hashing also takes $O(N)$, and selecting random indices needs $O(k)$. Therefore, the total time complexity of Random Selection is $O(2N + k)$.
>
> Considering the high-dimensional nature of embeddings, random selection typically has a much lower time complexity than smallest-magnitude selection.
>
> **For experiments:** To test time consumption, we use two popular embedding models: NV-Embed-v2 (based on Mistral-7B) and Stella (based on Qwen2-1.5B). We measure time for 2,000 generations, repeating the experiment 5 times to reduce randomness. Experiments run on Ubuntu 18.04 with an AMD EPYC 7Y83 64-Core CPU and a 4090 GPU.
>
> | **Model**         | **Model Size** | **Embedding Size** | **Inference Time (ms)** | **Smallest-magnitude Selection Time (ms)** | **Random Selection Time (ms)** |
> |-------------------|----------------|--------------------|-------------------------|--------------------------------------|--------------------------------|
> | **Stella**        | 1.5B           | 1024               | 4371.80 ± 204.80        | 716.30 ± 1.50                       | 31.49 ± 0.40                   |
> | **NV-Embed-v2**   | 7B              | 4096               | 13799.46 ± 459.30       | 3761.18 ± 276.59                    | 86.33 ± 0.49                   |
>
> **(3) Below is the comparison of watermark performance using smallest-magnitude and random selection.** We report p-value, ∆cos, and ∆l2 for detection capability, and cosine similarity to assess embedding quality. We set $K = 50$. The other parameters are the same as in the paper.
>
> | Dataset     | Method   | p-value ↓      | ∆cos(%) ↑        | ∆l2(%) ↓         | cos(%) w/o ↑ |
> |-------------|----------|---------------|------------------|------------------|-----------|
> | SST2        | Minimum  | $10^{-11}$    | 65.11            | -130.23          | 99.19    |
> |             | Random   | $10^{-11}$    | 72.81            | -145.62          | 98.87    |
> | MIND        | Minimum  | $10^{-11}$    | 72.14            | -144.28           | 99.23    |
> |             | Random   | $10^{-11}$    | 77.27            | -154.55          | 98.69    |
> | AGNews      | Minimum  | $10^{-10}$    | 21.83            | -43.65           | 99.27    |
> |             | Random   | $10^{-11}$    | 53.13            | -106.27          | 98.97    |
> | Enron Spam  | Minimum  | $10^{-10}$    | 47.75            | -95.5            | 99.21    |
> |             | Random   | $10^{-11}$    | 68.38            | -136.75          | 98.92    |
>
> **In summary, above analyses and experiments persent that both smallest-magnitude selection and random selection have their irreplaceable advantages and suited to their respective application scenarios:**
> * **Smallest-magnitude selection significantly benefits preserving embedding quality, with modifications to clean embeddings under 1%.** This is crucial for real-world scenarios where organizations aim to achieve higher rankings on leaderboards to promote their products while protecting their copyright.
> * **Random selection, in contrast, though sacrificing more embedding quality, saves considerable time**, making it more suitable for product deployment.
>
> We believe that **both approaches are meaningful**, and users can choose between them based on their specific application scenarios. We have included these analyses and experiment results in revised paper (Line 859). We're open to further feedback.

---

> ### Author Response · Authors · 2024-11-27
>
> Dear Reviewer nW4C,
>
> Thank you again for your time. As the deadline for discussion is approaching, we do wish to hear from you to see if our response resolves your concerns. We are happy to provide any additional clarifications if needed.

---

### Comment · Area_Chair_MoBF · 2024-11-23

Dear Reviewers,
The authors have responded to your valuable comments.
Please take a look at them!

Best,
AC

---

### Meta-Review · Area_Chair_MoBF · 2024-12-18

**Metareview:**

In this work, the authors proposed copyright protection of LLM-based EaaS via Embedding-Specific Watermark. This study is motivated from the authors' claim that ``... it is crucial to inject robust watermarks resistant to watermark removal attacks.'' and observation that ``...Existing watermarking methods typically inject a target embedding into embeddings through linear interpolation when the text contains triggers. However, this mechanism results in each watermarked embedding having the same component, which makes the watermark easy to identify and eliminate.''

During the rebuttal period, only one reviewer among a total of four has responded to authors' responses.
In particular, the concern of high computation cost, raised by at least two reviewers, has been properly addressed by the authors in terms of Smallest-magnitude vs. Random Selection together with time-complexity analysis.

The main weakness of this work is insufficient robustness evaluation.
Initially, the authors were not aware of this critical issue, and did not actively provide comprehensive robustness evaluations.
Although the authors added more experiments during the rebuttal period, it is a pity that the reviewers may not have the chance to review them. More importantly, it is needed to conduct robustness evaluation as a whole instead of just adding more experiments. For those that were not considered for experiments, it is hard to conclude the robustness of proposed framework.

**Additional Comments On Reviewer Discussion:**

Computation cost and robustness evaluation are two issues that were mainly discussed during the rebuttal period. As commented by Reviewer 3mUw, (s)he was still unsatisfactory about the robustness evaluation.

---

### Decision · Program_Chairs · 2025-01-22

Reject